# Stability & Generalisation of Gradient Descent for Shallow Neural Networks without the Neural Tangent Kernel

**Dominic Richards**
Department of Statistics
University of Oxford
24-29 St Giles', Oxford, OX1 3LB
Dominic.Richards94@gmail.com

**Ilja Kuzborskij**
DeepMind
London

iljak@deepmind.com

## Abstract

We revisit on-average algorithmic stability of Gradient Descent (GD) for training overparameterised shallow neural networks and prove new generalisation and excess risk bounds without the Neural Tangent Kernel (NTK) or Polyak-Łojasiewicz (PL) assumptions. In particular, we show oracle type bounds which reveal that the generalisation and excess risk of GD is controlled by an interpolating network with the shortest GD path from initialisation (in a sense, an interpolating network with the smallest relative norm). While this was known for kernelised interpolants, our proof applies directly to networks trained by GD without intermediate kernelisation. At the same time, by relaxing oracle inequalities developed here we recover existing NTK-based risk bounds in a straightforward way, which demonstrates that our analysis is tighter. Finally, unlike most of the NTK-based analyses we focus on regression with label noise and show that GD with early stopping is *consistent*.

## 1 Introduction

In a canonical statistical learning problem the learner is given a tuple of independently sampled training examples $S = (z_1, \ldots, z_n)$, where each example $z_i = (\mathbf{x}_i, y_i)$ consists of an *input* $\mathbf{x}_i$ and *label* $y_i$ jointly distributed according to some unknown probability measure $P$. In the following we assume that inputs belong to an Euclidean ball $\mathcal{B}_2^d(C_x)$ of radius $C_x$ and labels belong to $[-C_y, C_y]$. Based on training examples the goal of the learner is to select *parameters* $\mathbf{W}$ from some parameter space $\mathcal{W}$ in order to minimise the *statistical risk*

$$\mathcal{L}(\mathbf{W}) \stackrel{\text{def}}{=} \frac{1}{2} \int (f_{\mathbf{W}}(\mathbf{x}) - y)^2 \, \mathrm{d}P$$

where $f_{\mathbf{W}}$ is a predictor parameterised by $\mathbf{W}$. The best possible predictor in this setting is the *regression function* $f^\star$, which is defined as $f^\star(\mathbf{x}) = \int y \, \mathrm{d}P_{Y|\mathbf{X}=\mathbf{x}}$, while the minimum possible risk is equal to the noise-rate of the problem, which is given by $\sigma^2 = \int_{\mathcal{Z}} (f^\star(\mathbf{x}) - y)^2 \, \mathrm{d}P$.

In this paper we will focus on a *shallow neural network* predictor that takes the form

$$f_{\mathbf{W}}(\mathbf{x}) \stackrel{\text{def}}{=} \sum_{k=1}^{m} u_k \phi\left(\langle \mathbf{W}_k, \mathbf{x} \rangle\right) \qquad \left(\mathbf{x} \in \mathbb{R}^d, \ \mathbf{W} \in \mathbb{R}^{d \times m}\right)$$

defined with respect to some *activation function* $\phi : \mathbb{R} \to \mathbb{R}$, fixed *output layer* $\mathbf{u} \in \{\pm 1/\sqrt{m}\}^m$, and a tunable (possibly randomised) *hidden layer* $\mathbf{W}$. In particular, we will consider $f_{\mathbf{W}_T}$, where the

35th Conference on Neural Information Processing Systems (NeurIPS 2021).

hidden layer is obtained by minimising an empirical proxy of $\mathcal{L}$ called the *empirical risk* $\mathcal{L}_S(\mathbf{W}) \overset{\text{def}}{=} (2n)^{-1} \sum_{i=1}^{n} (f_{\mathbf{W}}(\mathbf{x}_i) - y_i)^2$ by running a *Gradient Descent (GD)* procedure: For $t \in [T]$ steps with initial parameters $\mathbf{W}_0$ and a step size $\eta > 0$, we have iterates $\mathbf{W}_{t+1} = \mathbf{W}_t - \eta \nabla \mathcal{L}_t(\mathbf{W}_t)$ where $\nabla \mathcal{L}_t(\mathbf{W})$ is the first order derivative of $\mathcal{L}_t(\mathbf{W})$.

Understanding the behaviour of the statistical risk for neural networks has been a long-standing topic of interest in the statistical learning theory [Anthony and Bartlett, 1999]. The standard approach to this problem is based on *uniform* bounds on the *generalisation gap*

$$\epsilon^{\text{Gen}}(\mathbf{W}_T) \overset{\text{def}}{=} \mathcal{L}(\mathbf{W}_T) - \mathcal{L}_S(\mathbf{W}_T) \, ,$$

which, given a parameter space $\mathcal{W}$, involves controlling the gap for the worst possible choice of $\mathbf{W} \in \mathcal{W}$ under some unknown data distribution. The theory then typically leads to the capacity-based (Rademacher complexity, VC-dimension, or metric-entropy based) bounds which hold with high probability (w.h.p.) over $S$ [Bartlett and Mendelson, 2002, Golowich et al., 2018, Neyshabur et al., 2018]: [1]

$$\sup_{\mathbf{W} \in \mathcal{W}} |\epsilon^{\text{Gen}}(\mathbf{W})| \lesssim \sqrt{\frac{\text{capacity}(\mathcal{W})}{n}} \, . \tag{1}$$

Thus, if one could simultaneously control the empirical risk and the capacity of the class of neural networks, one could control the statistical risk. Unfortunately, controlling the empirical risk $\mathcal{L}_T$ turns out to be a challenging part here since it is non-convex, and thus, it is not clear whether GD can minimise it up to a desired precision.

This issue has attracted considerable attention in recent years with numerous works [Du et al., 2018, Lee et al., 2019, Allen-Zhu et al., 2019, Oymak and Soltanolkotabi, 2020] demonstrating that *overparameterised* shallow networks (in a sense $m \gtrsim \text{poly}(n)$) trained on subgaussian inputs converge to global minima exponentially fast, namely, $\mathcal{L}_S(\mathbf{W}_T) \lesssim (1 - \eta \cdot {}^d/n)^T$. Loosely speaking, these proofs are based on the idea that a sufficiently overparameterised network trained by GD initialised at $\mathbf{W}_0$ with Gaussian entries, predicts closely to a solution of a Kernelised Least-Squares (KLS) formulation minimised by GD, where the kernel function called the *Neural Tangent Kernel (NTK)* [Jacot et al., 2018] is implicitly given by the activation function. This connection explains the observed exponential rate in case of shallow neural networks: For the NTK kernel matrix $\mathbf{K}$ the convergence rate of GD is $(1 - \eta \lambda_{\min}(\mathbf{K}))^T$, where $\lambda_{\min}(\mathbf{K})$ is its smallest eigenvalue, and as it turns out, for subgaussian inputs $\lambda_{\min}(\mathbf{K}) \gtrsim d/n$ [Bartlett et al., 2021]. [2] Naturally, the convergence was exploited to state bounds on the statistical risk: Arora et al. [2019] showed that for noise-free regression ($\sigma^2 = 0$) when $T \gtrsim 1/(\eta \lambda_{\min}(\mathbf{K}))$,

$$\mathcal{L}(\mathbf{W}_T) \lesssim \sqrt{\frac{\langle \mathbf{y}, (n\mathbf{K})^{-1} \mathbf{y} \rangle}{n}} \, . \tag{2}$$

Clearly, the bound is non-vacuous whenever $\langle \mathbf{y}, \mathbf{K}^{-1} \mathbf{y} \rangle \lesssim n^{2\alpha}$ for some $\alpha \in [0, 1)$, and Arora et al. [2019] present several examples of smooth target functions which satisfy this. More generally, for $f^\star$ which belongs to the Reproducing kernel Hilbert space (RKHS) induced by NTK one has $\langle \mathbf{y}, \mathbf{K}^{-1} \mathbf{y} \rangle \leq \|f^\star\|_{\mathcal{H}_{\text{NTK}}}^2$ [Schölkopf and Smola, 2002]. The norm-based control of the risk is standard in the literature on kernels, and thus, one might wonder to which extent neural networks are kernel methods in disguise? At the same time, some experimental and theoretical evidence [Bai and Lee, 2019, Seleznova and Kutyniok, 2020, Suzuki and Akiyama, 2021] suggest that connection to kernels might be good only at explaining the behaviour of very wide networks, much more overparameterised than those used in practice. Therefore, an interesting possibility is to develop alternative ways to analyse generalisation in neural networks: Is there a more straightforward kernel-free optimisation-based perspective?

In this paper we take a step in this direction and explore a kernel-free approach, which at the same time avoids worst-case type uniform bounds such as Eq. (1). In particular, we focus on the notion of the *algorithmic stability*: If an algorithm is insensitive to replacement (or removal) of an observation

---

[1]Throughout this paper, we use $f \lesssim g$ to say that there exists a universal constant $c > 0$ and some $k \in \mathbb{N}$ such that $f \leq cg \log^k(g)$ holds uniformly over all arguments.

[2]Which is a tightest known bound, Oymak and Soltanolkotabi [2020] prove a looser bound without distributional assumption on the inputs.

in a training tuple, then it must have a small generalisation gap. Thus, a natural question is whether GD is sufficiently stable when training overparameterised neural networks.

The stability of GD (and its stochastic counterpart) when minimising convex and non-convex smooth objective functions was first explored by Hardt et al. [2016]. Specifically, for a time-dependent choice of a step size $\eta_t = 1/t$ and a problem-dependent constant $\alpha \in (0, 1)$ they show that

$$\mathbf{E}\left[\epsilon^{\mathrm{Gen}}(\mathbf{W}_T) \mid \mathbf{W}_0, \mathbf{u}\right] \lesssim \ln(T) n^{-\alpha}.$$

Unfortunately, when combined with the NTK-based convergence rate of the empirical risk we have a vacuous bound since $\mathcal{L}_S(\mathbf{W}_T) \lesssim 1$. [3] This is because the stability is enforced through a quickly decaying step size rather than by exploiting a finer structure of the loss, and turns out to be insufficient to guarantee the convergence of the empirical risk. That said, several works Charles and Papailiopoulos [2018], Lei and Ying [2021] have proved stability bounds exploiting an additional structure in mildly non-convex losses. Specifically, they studied the stability of GD minimising a *gradient-dominated* empirical risk, meaning that for all $\mathbf{W}$ in some neighbourhood $\mathcal{W}$ and a problem-dependent quantity $\mu$, it is assumed that $\mathcal{L}_S(\mathbf{W}) - \min_{\mathbf{W}'} \mathcal{L}_S(\mathbf{W}') \leq \|\nabla \mathcal{L}_S(\mathbf{W})\|^2 / (2\mu)$. Having iterates of GD within $\mathcal{W}$ and assuming that the gradient is $\rho$-Lipschitz, this allows to show

$$\mathbf{E}\left[\epsilon^{\mathrm{Gen}}(\mathbf{W}_T) \mid \mathbf{W}_0, \mathbf{u}\right] \lesssim \frac{\rho}{\mu} \cdot \frac{1}{n}.$$

As it turns out, the condition is satisfied for the iterates of GD training overparameterised networks [Du et al., 2018] with high probability over $\mathbf{W}_0$. The key quantity that controls the bound is $\rho/\mu \lesssim 1/\lambda_{\min}(\mathbf{K})$ which can be interpreted as a *condition number* of the NTK matrix. However, it is known that for subgaussian inputs the condition number behaves as $n/d$ which renders the bound vacuous [Bartlett et al., 2021].

## 2   Our Contributions

In this paper we revisit algorithmic stability of GD for training overparameterised shallow neural networks, and prove new risk bounds **without the Neural Tangent Kernel (NTK) or Polyak-Łojasiewicz (PL)** machinery. In particular, we first show a bound on the generalisation gap and then specialise it to state risk bounds for regression with and without label noise. In the case of learning with noise we demonstrate that GD with a form of early stopping is *consistent*, meaning that the risk asymptotically converges to the noise rate $\sigma^2$.

Our analysis brings out a key quantity, which controls all our bounds, the *Regularised Empirical Risk Minimisation (R-ERM) Oracle* defined as

$$\Delta_S^{\mathrm{oracle}} \overset{\mathrm{def}}{=} \min_{\mathbf{W} \in \mathbb{R}^{d \times m}} \mathcal{L}_S(\mathbf{W}) + \mathcal{O}\left(\frac{\|\mathbf{W} - \mathbf{W}_0\|_F^2}{\eta T}\right) \quad \text{as} \quad \eta T \to \infty, \tag{3}$$

which means that $\Delta_S^{\mathrm{oracle}}$ is essentially an empirical risk of solution closest to initialisation (an interpolant when for $m$ large enough). We first consider a bound on the generalisation gap.

### 2.1   Generalisation Gap

For simplicity of presentation in the following we assume that the activation function $\phi$, its first and second derivatives are bounded. Assuming parameterisation $m \gtrsim (\eta T)^5$ we show (Corollary 1) that the expected generalisation gap is bounded as

$$\mathbf{E}\left[\epsilon^{\mathrm{Gen}}(\mathbf{W}_T) \mid \mathbf{W}_0, \mathbf{u}\right] \leq C \cdot \frac{\eta T}{n}\left(1 + \frac{\eta T}{n}\right) \mathbf{E}\left[\Delta_S^{\mathrm{oracle}} \mid \mathbf{W}_0, \mathbf{u}\right], \tag{4}$$

where $C$ is a constant independent from $n, T, \eta$.

---

[3]If $\eta_s = \frac{1}{s}$ we have $\mathcal{L}_S(\mathbf{W}_T) \lesssim \exp(\mu \sum_{j=1}^{T} \frac{1}{j}) \approx \frac{1}{T^\mu}$, thus, if $\mu \approx \frac{1}{n}$ we then require $T \sim \epsilon^{-n}$ for $\mathcal{L}_S(\mathbf{W}_T) \lesssim \epsilon$. Plugging this into the Generalisation Error bound we get that $\log(T) n^{-\alpha} = n^{1-\alpha} \log(1/\epsilon)$ which is vacuous as $n$ grows.

Dependence of $m$ on the total number of steps $T$ might appear strange at first, however things clear out once we pay attention to the scaling of the bound. Setting the step size $\eta$ to be constant, $T = n^\alpha$, and overparameterisation $m \gtrsim n^{5\alpha}$ for some free parameter $\alpha \in (0, 1]$ we have

$$\mathbf{E}\left[\epsilon^{\mathrm{Gen}}(\mathbf{W}_T) \mid \mathbf{W}_0, \mathbf{u}\right] = \mathcal{O}\left(\frac{1}{n} \mathbf{E}\left[\|\hat{\mathbf{W}} - \mathbf{W}_0\|_F^2 \mid \mathbf{W}_0, \mathbf{u}\right]\right) \quad \text{as} \quad n \to \infty,$$

where $\hat{\mathbf{W}}$ is chosen as parameters of a *minimal-norm interpolating network*, in a sense $\hat{\mathbf{W}} \in \mathrm{argmin}_{\mathbf{W} \in \mathbb{R}^{d \times m}}\{\|\mathbf{W} - \mathbf{W}_0\|_F^2 \mid \mathcal{L}_S(\mathbf{W}) = 0\}$. Thus, as Eq. (4) suggests, the generalisation gap is controlled by a minimal relative norm of an interpolating network. In comparison, previous work in the NTK setting obtained results of a similar type where in place of $\hat{\mathbf{W}}$ one has an interpolating minimal-norm solution to an NTK least-squares problem. Specifically, in Eq. (2) the generalisation gap is controlled by $\langle \mathbf{y}, (n\mathbf{K})^{-1}\mathbf{y}\rangle$, which is a squared norm of such solution.

The generalisation gap of course only tells us a partial story, since the behaviour of the empirical risk is unknown. Next, we take care of this and present bounds *excess risk* bounds.

## 2.2   Risk Bound without Label Noise

We first present a bound on the statistical risk which does not use the usual NTK arguments to control the empirical risk. For now assume that $\sigma^2 = 0$ meaning that there is no label noise and randomness is only in the inputs.

**NTK-free Risk Bound.**   In Corollary 2 we show that the risk is bounded as

$$\mathbf{E}\left[\mathcal{L}(\mathbf{W}_T) \mid \mathbf{W}_0, \mathbf{u}\right] \leq \left(1 + C \cdot \frac{\eta T}{n}\left(1 + \frac{\eta T}{n}\right)\right)\mathbf{E}\left[\Delta_S^{\mathrm{oracle}} \mid \mathbf{W}_0, \mathbf{u}\right]. \qquad (5)$$

Note that this bound looks very similar compared to the bound on generalisation gap. The difference lies in a "1+" term which accounts for the fact that $\mathcal{L}(\mathbf{W}_T) \leq \Delta_S^{\mathrm{oracle}}$ as we show in Lemma 2.

As before we let the step size be constant, set $T = n^\alpha$, and let overparameterisation be $m \gtrsim n^{5\alpha}$ for some $\alpha \in (0, 1]$. Then our risk bound implies that

$$\mathbf{E}\left[\mathcal{L}(\mathbf{W}_T) \mid \mathbf{W}_0, \mathbf{u}\right] = \mathcal{O}\left(\frac{1}{n^\alpha} \mathbf{E}\left[\|\hat{\mathbf{W}} - \mathbf{W}_0\|_F^2 \mid \mathbf{W}_0, \mathbf{u}\right]\right) \quad \text{as} \quad n \to \infty,$$

which comes as a simple consequence of bounding $\Delta_S^{\mathrm{oracle}}$ while choosing $\hat{\mathbf{W}}$ to be parameters of a minimal relative norm interpolating network as before. Recall that $y_i = f^\star(\mathbf{x}_i)$: The bound suggests that target functions $f^\star$ are only learnable if $n^{-\alpha} \mathbf{E}[\|\hat{\mathbf{W}} - \mathbf{W}_0\|_F^2 \mid \mathbf{W}_0, \mathbf{u}] \to 0$ as $n \to \infty$ for a fixed $\alpha$ and input distribution. A natural question is whether a class of such functions is non-empty. Indeed, the NTK theory suggests that it is not and in the following we will recover NTK-based results by relaxing our oracle inequality.

Similarly as in Section 2.1 we choose $T = n^\alpha$ and so we require $m \gtrsim n^{5\alpha}$, which trades off overparameterisation and convergence rate through the choice of $\alpha \in (0, 1]$. To the best of our knowledge this is the first result of this type, where one can achieve a smaller overparameterisation compared to the NTK literature at the expense of a slower convergence rate of the risk. Finally, unlike the NTK setting we do not require randomisation of the initialisation and the risk bound we obtain holds for any $\mathbf{W}_0$. [4]

**Comparison to NTK-based Risk Bound.**   Our generalisation bound can be naturally relaxed to obtain NTK-based empirical risk convergence rates. In this case we observe that our analysis is general enough to recover results of Arora et al. [2019] (up to a difference that our bounds hold in expectation rather than in high probability).

By looking at Eq. (5), our task is in controlling $\Delta_S^{\mathrm{oracle}}$: by its definition in Eq. (3) we can see that it can be bounded by the regularised empirical risk of any model, and we choose a Moore-Penrose pseudo-inverse solution to the least-squares problem supplied with the NTK feature map $\mathbf{x} \mapsto (\nabla_{\mathbf{W}} f_{\mathbf{W}}(\mathbf{x}))(\mathbf{W}_0)$. Here, entries of $\mathbf{W}_0$ are drawn from $\mathcal{N}(0, \nu_{\mathrm{init}}^2)$ for some appropriately

---

[4]In the NTK literature randomisation of $\mathbf{W}_0$ is required to guarantee that $\lambda_{\min}(\mathbf{K}) > 0$ [Du et al., 2018].

chosen $\nu_{\mathrm{init}}^2$, and entries of the outer layer $\mathbf{u}$ are distributed according to the uniform distribution over $\{\pm^1/\sqrt{m}\}$, independently from each other and the data. It is not hard to see that aforementioned pseudo-inverse solution $\mathbf{W}^{\mathrm{pinv}}$ will have zero empirical risk, and so we are left with controlling $\|\mathbf{W}^{\mathrm{pinv}} - \mathbf{W}_0\|_F^2$ as can be seen from the definition of $\Delta_S^{\mathrm{oracle}}$. Note that it is straightforward to do since $\mathbf{W}^{\mathrm{pinv}}$ has an analytic form. In Theorem 2 we carry out these steps and show that with high probability over initialisation $(\mathbf{W}_0, \mathbf{u})$,

$$\Delta_S^{\mathrm{oracle}} = \tilde{\mathcal{O}}_P \left( \frac{1}{n} \left\langle \mathbf{y}, (n\mathbf{K})^{-1}\mathbf{y} \right\rangle \right) \quad \text{as} \quad n \to \infty \,,$$

which combined with Eq. (5) recovers the result of Arora et al. [2019]. Note that we had to *relax* the oracle inequality, which suggests that the risk bound we give here is *tighter* compared to the NTK-based bound. Similarly, Arora et al. [2019] demonstrated that $\|\mathbf{W}_T - \mathbf{W}_0\|_F^2 = \tilde{\mathcal{O}}_P \left( \langle \mathbf{y}, (n\mathbf{K})^{-1}\mathbf{y} \rangle \right)$ holds w.h.p. over initialisation and ReLU activation function, however their proof requires a much more involved "coupling" argument where iterates $(\mathbf{W}_t)_{t=1}^T$ are shown to be close to the GD iterates minimising a KLS problem with NTK matrix.

## 2.3    Risk Bound with Label Noise and Consistency

So far we considered regression with random inputs, but without label noise. In this section we use our bounds to show that GD with early stopping is *consistent* in the regression with label noise. Here labels are generated as $y_i = f^\star(\mathbf{x}_i) + \varepsilon_i$ where zero-mean random variables $(\varepsilon_i)_{i=1}^n$ are i.i.d., almost surely *bounded*,[5] and $\mathbf{E}[\varepsilon_1^2] = \sigma^2$.

For now we will use the same settings of parameters as in Section 2.2: Recall that $\eta$ is constant, $T = n^\alpha, m \gtrsim n^{5\alpha}$ for a free parameter $\alpha \in (0, 1]$. In addition assume a *well-specified* scenario which means that $m$ is large enough such that for some subset of parameters $\mathcal{W}$ we achieve $\mathcal{L}(\mathbf{W}^\star) = \sigma^2$ for some parameters $\mathbf{W}^\star$. Employing our risk bound of Eq. (5), we relax R-ERM oracle as

$$\mathbf{E}[\Delta_S^{\mathrm{oracle}} \mid \mathbf{W}_0, \mathbf{u}] \lesssim \sigma^2 + n^{-\alpha}\|\mathbf{W}^\star - \mathbf{W}_0\|_F^2 \,, \quad \text{where} \quad \mathbf{W}^\star \in \mathrm{argmin}_{\mathbf{W} \in \mathbb{R}^{d \times m}} \mathcal{L}(\mathbf{W}) \,.$$

Then we immediately have an $\alpha$-dependent risk bound

$$\mathbf{E}\left[\mathcal{L}(\mathbf{W}_T) \mid \mathbf{W}_0, \mathbf{u}\right] \leq \left(1 + \frac{C}{n^{1-\alpha}}\right) \left(\sigma^2 + \mathcal{O}\left(\frac{\|\mathbf{W}^\star - \mathbf{W}_0\|_F^2}{n^\alpha}\right)\right) \quad \text{as} \quad n \to \infty \,. \quad (6)$$

It is important to note that for any tuning $\alpha \in (0, 1)$, as long as $\mathbf{W}^\star$ is constant in $n$ (which is the case in the parametric setting), we have $\mathbf{E}\left[\mathcal{L}(\mathbf{W}_T) \mid \mathbf{W}_0, \mathbf{u}\right] \to \sigma^2$ as $n \to \infty$, and so we achieve *consistency*. Adaptivity to the noise achieved here is due to *early stopping*: Indeed, recall that we take $T = n^\alpha$ steps and having smaller $\alpha$ mitigates the effect of the noise as can be seen from Eq. (6). This should come as no surprise as early stopping is well-known to have a regularising effect in GD methods [Yao et al., 2007]. The current literature on the NTK deals with this through a kernelisation perspective by introducing an explicit $L2$ regularisation [Hu et al., 2021], while risk bound of [Arora et al., 2019] designed for a noise-free setting would be vacuous in this case.

## 2.4    Future Directions and Limitations

We presented generalisation and risk bounds for shallow nets controlled by the Regularised Empirical Risk Minimisation oracle $\Delta_S^{\mathrm{oracle}}$. By straightforward relaxation of $\Delta_S^{\mathrm{oracle}}$, we showed that the risk can be controlled by the minimal (relative) norm of an interpolating network which is tighter than the NTK-based bounds. There are several interesting venues which can be explored based on our results. For example, by assuming a specific form of a target function $f^\star$ (e.g., a "teacher" network), one possibility is to characterise the minimal norm. This would allow us to understand better which such target function are *learnable* by GD.

One limitation of our analysis is smoothness of the activation function and its boundedness. While boundedness can be easily dealt with by localising the smoothness analysis (using a Taylor approximation around each GD iterate) and randomising $(\mathbf{W}_0, \mathbf{u})$, extending our analysis to non-smooth activations (such as ReLU $\phi(x) = \max(x, 0)$) appears to be non-trivial because our stability analysis

---

[5]We assume boundedness of labels throughout our analysis to ensure that $\mathcal{L}_S(\mathbf{W}_0)$ is bounded almost surely for simplicity. This can be relaxed by introducing randomised initialisation, e.g. as in [Du et al., 2018].

crucially relies on the control of the smallest eigenvalue of the Hessian. Another peculiarity of our analysis is a time-dependent parameterisation requirement $m \gtrsim (\eta T)^5$. Making parameterisation $n$-dependent introduces an implicit early-stopping condition, which helped us to deal with label noise. On the other hand, such requirement might be limiting when dealing with noiseless interpolating learning where one could require to have $\eta T \to \infty$ while keeping $n$ finite.

## Notation

In the following denote $\ell(\mathbf{w}, (\mathbf{x}, y)) \stackrel{\text{def}}{=} \frac{1}{2}(f_{\mathbf{W}}(\mathbf{x}) - y)^2$. Operator norm is denoted as $\|\cdot\|_{\text{op}}$, while $L_2$ norm is denoted by $\|\cdot\|_2$ or $\|\cdot\|_F$. We use notation $(a \vee b) \stackrel{\text{def}}{=} \max\{a, b\}$ and $(a \wedge b) \stackrel{\text{def}}{=} \min\{a, b\}$ throughout the paper. Let $(\mathbf{W}_t^{(i)})_t$ be the iterates of GD obtained from the data set with a resampled data point:

$$S^{(i)} \stackrel{\text{def}}{=} (z_1, \ldots, z_{i-1}, \widetilde{z}_i, z_{i+1}, \ldots, z_n)$$

where $\widetilde{z}_i$ is an independent copy of $z_i$. Moreover, denote a remove-one version of $S$ by

$$S^{\backslash i} \stackrel{\text{def}}{=} (z_1, \ldots, z_{i-1}, z_{i+1}, \ldots, z_n).$$

## 3 Main Result and Proof Sketch

This section presents both the main results as well as a proof sketch. Precisely, Section 3.1 presents the main result and Section 3.2 the proof sketch.

### 3.1 Main Results

We formally make the following assumption regarding the regularity of the activation function.

**Assumption 1** (Activation). *The activation $\phi(u)$ is continuous and twice differentiable with constant $B_\phi, B_{\phi'}, B_{\phi''} \geq 0$ bounding $|\phi(u)| \leq B_\phi$, $|\phi'(u)| \leq B_{\phi'}$ and $|\phi''(u)| \leq B_{\phi''}$ for any $u \in \mathbb{R}$.*

This is satisfied for sigmoid as well as hyperbolic tangent activations.

**Assumption 2** (Inputs, labels, and the loss function). *For constants $C_x, C_y, C_0 > 0$, inputs belong to $\mathcal{B}_2^d(C_x)$, labels belong to $[-C_y, C_y]$, and loss is uniformly bounded by $C_0$ almost surely.*

A consequence of the above is an essentially constant smoothness of the loss and the fact that the Hessian scales with $1/\sqrt{m}$ (see Appendix A for the proof). These facts will play a key role in our stability analysis. [6]

**Lemma 1** (Smoothness and curvature). *Fix $\mathbf{W}, \widetilde{\mathbf{W}} \in \mathbb{R}^{d \times m}$. Consider Assumption 1, Assumption 2, and assume that $\mathcal{L}_S(\widetilde{\mathbf{W}}) \leq C_0^2$. Then, for any $S$,*

$$\lambda_{\max}(\nabla^2 \mathcal{L}_S(\mathbf{W})) \leq \rho \quad where \quad \rho \stackrel{\text{def}}{=} C_x^2 \left( B_{\phi'}^2 + B_{\phi''} B_\phi + \frac{B_{\phi''} C_y}{\sqrt{m}} \right),$$

$$\min_{\alpha \in [0,1]} \lambda_{\min}(\nabla^2 \mathcal{L}_S(\widetilde{\mathbf{W}} + \alpha(\mathbf{W} - \widetilde{\mathbf{W}}))) \geq -\frac{B_{\phi''}(B_{\phi'} C_x + C_0)}{\sqrt{m}} \cdot (1 \vee \|\mathbf{W} - \widetilde{\mathbf{W}}\|_F).$$

Given these assumptions we are ready to present a bound on the Generalisation Error of the gradient descent iterates.

**Theorem 1** (Generalisation Error). *Consider Assumptions 1 and 2. Fix $t > 0$. If $\eta \leq 1/(2\rho)$ and*

$$m \geq 144(\eta t)^2 C_x^4 C_0^2 B_{\phi''}^2 \left( 4B_{\phi'} C_x \sqrt{\eta t} + \sqrt{2} \right)^2 \tag{7}$$

$$then \qquad \mathbf{E}\left[\epsilon^{Gen}(\mathbf{W}_{t+1}) \mid \mathbf{W}_0, \mathbf{u}\right] \leq b \left( \frac{\eta}{n} + \frac{\eta^2 t}{n^2} \right) \sum_{j=0}^{t} \mathbf{E}\left[\mathcal{L}_S(\mathbf{W}_j) \mid \mathbf{W}_0, \mathbf{u}\right]$$

*where $b = 16e^3 C_x^{\frac{3}{2}} B_{\phi'}^2 (1 + C_x^{\frac{3}{2}} B_{\phi'}^2)$.*

---

[6]We use notation $(a \vee b) \stackrel{\text{def}}{=} \max\{a, b\}$ and $(a \wedge b) \stackrel{\text{def}}{=} \min\{a, b\}$ throughout the paper.

*Proof.* For full proof see Appendix C with sketch proof in Section 3.2. $\qquad\square$

Theorem 1 then provides a formal upper bound on the Generalisation Gap of a shallow neural network trained with GD. Specifically, provided the network is sufficiently wide the generalisation gap can be controlled through both: the gradient descent trajectory evaluated at the Empirical Risk $\sum_{t=0}^{T} \mathbf{E}[\mathcal{L}_S(\mathbf{W}_t) \mid \mathbf{W}_0, \mathbf{u}]$, as well as step size and number of iterations in the multiplicative factor. Similar bounds for the test performance of gradient descent have been given by Lei and Ying [2020], although in our case the risk is non-convex, and thus, a more delicate bound on the Optimisation Gap is required. This is summarised within the lemma, which is presented in an "oracle" inequality manner.

**Lemma 2** (Optimisation Error). *Consider Assumptions 1 and 2. Fix $t > 0$. If $\eta \leq 1/(2\rho)$, then*

$$\frac{1}{t} \sum_{j=0}^{t} \mathcal{L}_S(\mathbf{W}_j) \leq \min_{\mathbf{W} \in \mathbb{R}^{d \times m}} \left\{ \mathcal{L}_S(\mathbf{W}) + \frac{\|\mathbf{W} - \mathbf{W}_0\|_F^2}{\eta t} + \frac{\widetilde{b}\|\mathbf{W} - \mathbf{W}_0\|_F^3}{\sqrt{m}} \right\} + \widetilde{b} C_0 \cdot \frac{(\eta t)^{\frac{3}{2}}}{\sqrt{m}}$$

*where $\widetilde{b} = C_x^2 B_{\phi''} (B_{\phi'} C_x + C_0)$.*

*Proof.* Full proof is given in Appendix B with sketch proof in Section 3.2. $\qquad\square$

By combining Theorem 1 and Lemma 2 we get the following where $\Delta_S^{\text{oracle}}$ is defined in Eq. (3):

**Corollary 1.** *Assume the same as in Theorem 1 and Lemma 2. Then,*

$$\mathbf{E}\left[\epsilon^{\text{Gen}}(\mathbf{W}_T) \mid \mathbf{W}_0, \mathbf{u}\right] \leq C \cdot \frac{\eta T}{n} \left(1 + \frac{\eta T}{n}\right) \mathbf{E}\left[\Delta_S^{\text{oracle}} \mid \mathbf{W}_0, \mathbf{u}\right]$$

*where $\Delta_S^{\text{oracle}}$ is defined in Eq. (3) and $C$ is a constant independent from $n, T, \eta$.*

The above combined with Lemma 2, and the fact that $\mathcal{L}_S(\mathbf{W}_T) = \min_{t \in [T]} \mathcal{L}_S(\mathbf{W}_t)$ gives us:

**Corollary 2.** *Assume the same as in Theorem 1 and Lemma 2. Then,*

$$\mathbf{E}\left[\mathcal{L}(\mathbf{W}_T) \mid \mathbf{W}_0, \mathbf{u}\right] \leq \left(1 + C \cdot \frac{\eta T}{n} \left(1 + \frac{\eta T}{n}\right)\right) \mathbf{E}\left[\Delta_S^{\text{oracle}} \mid \mathbf{W}_0, \mathbf{u}\right] .$$

*Proof.* The proof is given in Appendix E. $\qquad\square$

Finally, $\Delta_S^{\text{oracle}}$ is controlled by the norm of the NTK solution, which establishes connection to the NTK-based risk bounds:

**Theorem 2** (Connection between $\Delta_S^{\text{oracle}}$ and NTK). *Consider Assumption 1 and that $\eta T = n$. Moreover, assume that entries of $\mathbf{W}_0$ are i.i.d., that $\lambda_{\min}(\mathbf{K}) \gtrsim 1/n$, and assume that $\mathbf{u} \sim \text{unif}\left(\{\pm 1/\sqrt{m}\}\right)^m$ independently from all sources of randomness. Then, with probability least $1 - \delta$ for $\delta \in (0, 1)$, over $(\mathbf{W}_0, \mathbf{u})$,*

$$\Delta_S^{\text{oracle}} = \tilde{\mathcal{O}}_P \left(\frac{1}{n} \left\langle \mathbf{y}, (n\mathbf{K})^{-1}\mathbf{y} \right\rangle\right) \quad \text{as} \quad n \to \infty .$$

*Proof.* The proof is given in Appendix D. $\qquad\square$

### 3.2 Proof Sketch of Theorem 1 and Lemma 2

Throughout this sketch let expectation be understood as $\mathbf{E}[\cdot] = \mathbf{E}[\cdot \mid \mathbf{W}_0, \mathbf{u}]$. For brevity we will also vectorise parameter matrices, so that $\mathbf{W} \in \mathbb{R}^{dm}$ and thus $\|\cdot\|_2 = \|\cdot\|_F$.

Let us begin with the bound on the generalisation gap, for which we use the notion of algorithmic stability [Bousquet and Elisseeff, 2002]. With $\mathbf{W}_t^{(i)}$ denoting a GD iterate with the dataset with the resampled data point $S^{(i)}$, the Generalisation Gap can be rewritten [Shalev-Shwartz and Ben-David, 2014, Chap. 13]

$$\epsilon^{\text{Gen}} = \mathbf{E}[\mathcal{L}(\mathbf{W}_T) - \mathcal{L}_S(\mathbf{W}_T)] = \frac{1}{n} \sum_{i=1}^{n} \mathbf{E}[\ell(\mathbf{W}_T, \widetilde{z}_i) - \ell(\mathbf{W}_T^{(i)}, \widetilde{z}_i)] .$$

The equality suggests that if trained parameters do not vary much when a data point is resampled, that is $\mathbf{W}_T \approx \mathbf{W}_T^{(i)}$, then the Generalisation Gap will be small. Indeed, prior work [Hardt et al., 2016, Kuzborskij and Lampert, 2018] has been dedicated to proving uniform bounds on $L_2$ norm $\max_{i \in [n]} \|\mathbf{W}_T - \mathbf{W}_T^{(i)}\|_2$. In our case, we consider the more refined squared $L_2$ expected stability and consider a bound for smooth losses similar to that of Lei and Ying [2020]:

**Lemma 3.** *Consider Assumptions 1 and 2. Then,*

$$
\epsilon^{\mathrm{Gen}} \lesssim \sqrt{\mathbf{E}\left[\mathcal{L}_S(\mathbf{W}_T)\right]} \sqrt{\frac{1}{n}\sum_{i=1}^{n}\mathbf{E}\left[\|\mathbf{W}_T - \mathbf{W}_T^{(i)}\|_{op}^2\right] + \frac{1}{n}\sum_{i=1}^{n}\mathbf{E}\left[\|\mathbf{W}_T - \mathbf{W}_T^{(i)}\|_{op}^2\right]} \ .
$$

*Proof.* The proof is given in Appendix C.1. □

To this end, we bound $\|\mathbf{W}_T - \mathbf{W}_T^{(i)}\|_2^2$ recursively by using the definition of the gradient iterates and applying Young's inequality $(a+b)^2 \leq (1+\frac{1}{t})a^2 + (1+t)b^2$ to get that for any $i \in [n]$:

$$
\|\mathbf{W}_{t+1} - \mathbf{W}_{t+1}^{(i)}\|_2^2 \tag{8}
$$
$$
= \|\mathbf{W}_t - \mathbf{W}_t^{(i)} - \eta(\nabla\mathcal{L}_S(\mathbf{W}_t) - \nabla\mathcal{L}_{S^{(i)}}(\mathbf{W}_t^{(i)}))\|_2^2
$$
$$
= \|\mathbf{W}_t - \mathbf{W}_t^{(i)} - \eta(\nabla\mathcal{L}_{S\setminus i}(\mathbf{W}_t) - \nabla\mathcal{L}_{S\setminus i}(\mathbf{W}_t^{(i)})) + \frac{\eta}{n}(\nabla\ell(\mathbf{W}_t, z_i) - \nabla\ell(\mathbf{W}_t^{(i)}, \tilde{z}_i)\|_2^2
$$
$$
\leq (1 + \frac{1}{t}) \underbrace{\|\mathbf{W}_t - \mathbf{W}_t^{(i)} - \eta\left(\nabla\mathcal{L}_{S\setminus i}(\mathbf{W}_t) - \nabla\mathcal{L}_{S\setminus i}(\mathbf{W}_t^{(i)})\right)\|_2^2}_{\text{Expansiveness of the Gradient Update}}
$$
$$
+ \frac{\eta^2(1+t)}{n^2}\left(\|\nabla\ell(\mathbf{W}_t, z_i)\|_2^2 + \|\nabla\ell(\mathbf{W}_t^{(i)}, \tilde{z}_i)\|_2^2\right) \ . \tag{9}
$$

The analysis of the *expansiveness of the gradient update* then completes the recursive relationship [Hardt et al., 2016, Kuzborskij and Lampert, 2018, Lei and Ying, 2020, Richards and Rabbat, 2021]. Clearly, a trivial handling of the term (using smoothness) would yield a bound $(1 + \eta\rho)^2\|\mathbf{W}_t - \mathbf{W}_t^{(i)}\|_2^2$, which leads to an exponential blow-up when unrolling the recursion. So, we must ensure that the expansivity coefficient is no greater than one. While the classic analysis of Hardt et al. [2016] controls this by having a polynomially decaying learning rate schedule, here we leverage on observation of Richards and Rabbat [2021] that the negative Eigenvalues of the loss's Hessian $\nabla^2\mathcal{L}_{S\setminus i}(\cdot) \in \mathbb{R}^{dm \times dm}$ can control the expansiveness. This boils down to Lemma 1:

$$
\lambda_{\min}\left(\int_0^1 \nabla^2\mathcal{L}_{S\setminus i}(\mathbf{W}_t^{(i)} + \alpha(\mathbf{W}_t - \mathbf{W}_t^{(i)}))\,\mathrm{d}\alpha\right) \gtrsim -\frac{1 \vee \|\mathbf{W}_t - \mathbf{W}_t^{(i)}\|_2}{\sqrt{m}} \gtrsim -\sqrt{\frac{\eta t}{m}}
$$

where $\|\mathbf{W}_t - \mathbf{W}_t^{(i)}\|_2 \lesssim \sqrt{\eta t}$ comes from smoothness and standard analysis of GD (see Lemma 6). Note that the eigenvalue scales $1/\sqrt{m}$ because Hessian has a block-diagonal structure.

**Controlling expansiveness of GD updates.** Abbreviate $\Delta = \nabla\mathcal{L}_{S\setminus i}(\mathbf{W}_t) - \nabla\mathcal{L}_{S\setminus i}(\mathbf{W}_t^{(i)})$ and $\nabla^2 = \int_0^1 \nabla^2\mathcal{L}_{S\setminus i}(\mathbf{W}_t^{(i)} + \alpha(\mathbf{W}_t - \mathbf{W}_t^{(i)}))\,\mathrm{d}\alpha$. Now we "open" the squared norm

$$
\|\mathbf{W}_t - \mathbf{W}_t^{(i)} - \eta\Delta\|_2^2 = \|\mathbf{W}_t - \mathbf{W}_t^{(i)}\|_2^2 + \eta^2\|\Delta\|_2^2 - 2\eta\left\langle\mathbf{W}_t - \mathbf{W}_t^{(i)}, \Delta\right\rangle
$$

and use Taylor's theorem for gradients to have

$$
\eta^2\|\Delta\|_2^2 - 2\eta\left\langle\mathbf{W}_t - \mathbf{W}_t^{(i)}, \Delta\right\rangle = \left\langle\Delta, \left(\eta^2\nabla^2 - 2\eta\mathbf{I}\right)(\mathbf{W}_t - \mathbf{W}_t^{(i)})\right\rangle
$$
$$
= \left\langle\mathbf{W}_t - \mathbf{W}_t^{(i)}, \eta\nabla^2\left(\eta\nabla^2 - 2\mathbf{I}\right)(\mathbf{W}_t - \mathbf{W}_t^{(i)})\right\rangle
$$
$$
\lesssim \left(\eta\sqrt{\frac{\eta t}{m}} + \eta^2\frac{\eta t}{m}\right)\|\mathbf{W}_t - \mathbf{W}_t^{(i)}\|_2^2
$$

where we noted that $(\eta\nabla^2 - 2\mathbf{I})$ has only negative eigenvalues due to assumption $\eta \leq 1/(2\rho)$.

By rearranging this inequality one can verify that the gradient operator is *approximately co-coercive* [Hardt et al., 2016]. This result, crucial to our proof, is supported by some empirical evidence: In Fig. 1 we show a synthetic experiment where we train a shallow neural network with sigmoid activation. Our handling of expansiveness eventually gives

$$\|\mathbf{W}_t - \mathbf{W}_t^{(i)} - \eta\Delta\|_2^2 \lesssim \left(1 + \eta\sqrt{\frac{\eta t}{m}}\right) \|\mathbf{W}_t - \mathbf{W}_t^{(i)}\|_2^2 \,.$$

Plugging this back into Eq. (8)-(9) we unroll the recursion:

$$\|\mathbf{W}_{t+1} - \mathbf{W}_{t+1}^{(i)}\|_2^2 \lesssim \frac{\eta^2 t}{n^2} \sum_{j=0}^{t} \left(1 + \frac{1}{t}\right)^t \left(1 + \eta\sqrt{\frac{\eta t}{m}}\right)^t \left(\|\nabla\ell(\mathbf{W}_j, z_i)\|_2^2 + \|\nabla\ell(\mathbf{W}_j^{(i)}, \widetilde{z}_i)\|_2^2\right) \,.$$

The above suggest that to prevent exponential blow-up (and ensure that the expansiveness remains under control), it's enough to have $\eta\sqrt{\frac{\eta t}{m}} \leq 1/t$, which is an assumption of the theorem, enforcing relationship between $\eta t$ and $m$.

**On-average stability bound.** Now we are ready to go back to Lemma 3 and complete the proof of the generalisation bound. Since we have a squared loss we have $\|\nabla\ell(\mathbf{W}_j, z_i)\|_2^2 \lesssim \ell(\mathbf{W}_j, z_i)$, and thus, when summing over $i \in [n]$ we recover the Empirical risk $\frac{1}{n}\sum_{i=1}^{n} \|\nabla\ell(\mathbf{W}_j, z_i)\|_2^2 \lesssim \mathcal{L}_S(\mathbf{W}_j)$. Finally, noting that $\mathbf{E}[\ell(\mathbf{W}_j, z_i)] = \mathbf{E}[\ell(\mathbf{W}_j^{(i)}, \widetilde{z}_i)]$ we then arrive at the following bound on the expected squared $L_2$ stability

$$\frac{1}{n}\sum_{i=1}^{n} \mathbf{E}[\|\mathbf{W}_{t+1} - \mathbf{W}_{t+1}^{(i)}\|_2^2] \lesssim \frac{\eta^2 t}{n^2} \sum_{j=0}^{t} \mathbf{E}[\mathcal{L}_S(\mathbf{W}_j)] \,.$$

Note that the Generalisation Gap is directly controlled by the Optimisation Performance along the path of the iterates, matching the shape of bounds by Lei and Ying [2020]. Using smoothness of the loss as well as that $\mathcal{L}_S(\mathbf{W}_T) \leq \frac{1}{T}\sum_{j=0}^{T} \mathcal{L}_S(\mathbf{W}_j)$, the risk is bounded when $m \gtrsim (\eta t)^3$ as

$$\mathbf{E}[\mathcal{L}(\mathbf{W}_T)] \lesssim \left(1 + \frac{\eta T}{n}\left(1 + \frac{\eta T}{n}\right)\right) \frac{1}{T}\sum_{j=0}^{T} \mathbf{E}[\mathcal{L}_S(\mathbf{W}_j)] \,.$$

**Optimisation error.** We must now bound the Optimisation Error averaged across the iterates. Following standard optimisation arguments for gradient descent on a smooth objective we then get

$$\mathcal{L}_S(\mathbf{W}_{t+1}) \leq \mathcal{L}_S(\mathbf{W}_t) - \frac{\eta}{2}\|\nabla\mathcal{L}_S(\mathbf{W}_t)\|_2^2$$

At this point convexity would usually be applied to upper bound $\mathcal{L}_S(\mathbf{W}_t) \leq \mathcal{L}_S(\hat{\mathbf{W}}) - \langle\nabla\mathcal{L}_S(\mathbf{W}_t), \hat{\mathbf{W}} - \mathbf{W}_t\rangle$ where $\hat{\mathbf{W}}$ is a minimiser of $\mathcal{L}_S(\cdot)$. The analysis differs from this approach in two respects. Firstly, we do not consider a minimiser of $\mathcal{L}_S(\cdot)$ but a general point $\mathbf{W}$, which is then optimised over at the end. Secondly, the objective is not convex, and therefore, we leverage that the Hessian negative Eigenvalues are on the order of $-\frac{1}{\sqrt{m}}$ to arrive at the upper bound (Lemma 1):

$$\mathcal{L}_S(\mathbf{W}_t) \lesssim \underbrace{\mathcal{L}_S(\mathbf{W}) - \langle\nabla\mathcal{L}_S(\mathbf{W}_t), \mathbf{W} - \mathbf{W}_t\rangle}_{\text{Convex Component}} + \underbrace{\frac{1}{\sqrt{m}}\|\mathbf{W} - \mathbf{W}_t\|_2^3}_{\text{Negative Hessian Eigenvalues}}$$

After this point, the standard optimisation analysis for gradient descent can be performed i.e. $-\langle\nabla\mathcal{L}_S(\mathbf{W}_t), \mathbf{W} - \mathbf{W}_t\rangle - \frac{\eta}{2}\|\mathcal{L}_S(\mathbf{W}_t)\|_2^2 = \frac{1}{\eta}\left(\|\mathbf{W} - \mathbf{W}_t\|_2^2 - \|\mathbf{W} - \mathbf{W}_{t+1}\|_2^2\right)$, which then yields to a telescoping sum over $t = 0, 1, \ldots, T-1$. This leads to the upper bound for $\mathbf{W} \in \mathbb{R}^{m \times d}$

$$\frac{1}{T}\sum_{j=0}^{T} \mathbf{E}[\mathcal{L}_S(\mathbf{W}_j)] \lesssim \mathcal{L}_S(\mathbf{W}) + \frac{\|\mathbf{W} - \mathbf{W}_0\|_2^2}{\eta T} + \frac{1}{\sqrt{m}} \cdot \frac{1}{T}\sum_{j=0}^{T} \|\mathbf{W} - \mathbf{W}_j\|_2^3$$

where we plug this into the generalisation bound and take a minimum over $\mathbf{W} \in \mathbb{R}^{md}$.

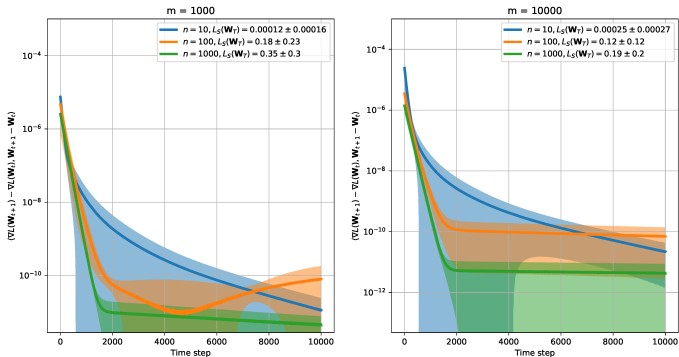

Figure 1: The gradient operator of an overparameterised shallow neural network is almost-monotone: Here, inputs are uniformly distributed on a 10-sphere and labels are generated by $\mathbf{x} \mapsto \frac{e^{\langle \mathbf{W}^\star, \mathbf{x} \rangle}}{1 + e^{\langle \mathbf{W}^\star, \mathbf{x} \rangle}}$ where $\mathbf{W}^\star$ is once sampled from $\mathcal{N}(\mathbf{0}, \mathbf{I}_{10})$ at the beginning of the training. The initialisation $\mathbf{W}_0$ is sampled from $\mathcal{N}(\mathbf{0}, \mathbf{I}_{10})$ and each experiment is performed 10 times (which is reflected by standard deviation). The shallow network is then trained by GD with $\eta = 1$, $T = 10^4$, and sigmoid activation.

## 4 Additional Related Literature

**Stability of Gradient Descent**. Algorithmic stability [Bousquet and Elisseeff, 2002] has been used to investigate the generalisation performance of GD in a number of works [Hardt et al., 2016, Kuzborskij and Lampert, 2018, Chen et al., 2018, Lei and Ying, 2020, Richards and Rabbat, 2021]. While near optimal bounds are achieved for convex losses, the non-convex case aligning with this work is more challenging. Within [Hardt et al., 2016, Kuzborskij and Lampert, 2018] in particular a restrictive $1/t$ step size is required if $t$ iterations of gradient descent are performed. This was partially alleviated within [Richards and Rabbat, 2021] which demonstrated that the magnitude of Hessian's negative eigenvalues can be leveraged to allow for much larger step sizes to be taken. In this work we also utilise the magnitude of Hessian's negative Eigenvalue, although a more delicate analysis is required as both: the gradient is potentially unbounded in our case; and the Hessian's negative eigenvalues need to be bounded when evaluated across multiple points. We note the scaling of the Hessian network has also previously been observed within [Liu et al., 2020a,b], with empirical investigations into the magnitude of the Hessian's negative Eigenvalues conducted within [Sagun et al., 2016, Yuan et al., 2019]. Stability-based generalisation bounds were also shown for certain types of non-convex functions, such Polyak-Łojasiewicz functions [Charles and Papailiopoulos, 2018, Lei and Ying, 2021] and functions with strict saddles among stationary points [Gonen and Shalev-Shwartz, 2017]. In line with our results, Rangamani et al. [2020] shows that for interpolating kernel predictors, minimizing the norm of the ERM solution minimizes stability.

Several works have also recently demonstrated consistency of GD with early stopping for training shallow neural networks in the presence of label noise. The concurrent work [Ji et al., 2021] also showed that shallow neural networks trained with gradient descent are consistent, although their approach distinctly different to ours e.g. leveraging structure of logistic loss as well as connections between shallow neural networks and random feature models / NTK. Earlier, Li et al. [2020] showed consistency under a certain gaussian-mixture type parametric classification model. In a more general nonparametric setting, Kuzborskij and Szepesvári [2021] showed that early-stopped GD is consistent when learning Lipschitz regression functions.

Certain notions of stability, such the uniform stability (taking sup over the data rather than expectation) allows to prove **high-probability risk bounds (h.p.)** [Bousquet and Elisseeff, 2002] which are known to be optimal up to log-terms [Feldman and Vondrak, 2019, Bousquet et al., 2020]. In this work we show bounds in expectation, or equivalently, control the first moment of a generalisation gap. To prove a h.p. bounds while enjoying benefits of a stability in expectation, one would need to control higher order moments [Maurer, 2017, Abou-Moustafa and Szepesvári, 2019], however often this is done through a higher-order uniform stability: Since our proof closely relies on the stability in expectation, it is not clear whether our results can be trivially stated with high probability.

## Acknowledgements

D.R. is supported by the EPSRC and MRC through the OxWaSP CDT programme (EP/L016710/1), and the London Mathematical Society ECF-1920-61.

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
