## Notation

In the following denote $\ell(\mathbf{w}, (\mathbf{x}, y)) \stackrel{\text{def}}{=} \frac{1}{2}(f_{\mathbf{W}}(\mathbf{x}) - y)^2$. Unless stated otherwise, we work with vectorised quantities so $\mathbf{W} \in \mathbb{R}^{dm}$ and therefore simply interchange $\|\cdot\|_2$ with $\|\cdot\|_F$. We also use notation $(\mathbf{W})_k$ so select $k$-th block of size $d$, that is $(\mathbf{W})_k = [W_{(d-1)k+1}, \ldots, W_{dk}]^\top$. We use notation $(a \vee b) \stackrel{\text{def}}{=} \max\{a, b\}$ and $(a \wedge b) \stackrel{\text{def}}{=} \min\{a, b\}$ throughout the paper. Let $(\mathbf{W}_t^{(i)})_t$ be the iterates of GD obtained from the data set with a resampled data point:

$$S^{(i)} \stackrel{\text{def}}{=} (z_1, \ldots, z_{i-1}, \widetilde{z}_i, z_{i+1}, \ldots, z_n)$$

where $\widetilde{z}_i$ is an independent copy of $z_i$. Moreover, denote a remove-one version of $S$ by

$$S^{\backslash i} \stackrel{\text{def}}{=} (z_1, \ldots, z_{i-1}, z_{i+1}, \ldots, z_n).$$

## A   Smoothness and Curvature of the Empirical Risk (Proof of Lemma 1)

**Lemma 1 (restated).** *Fix* $\mathbf{W}, \widetilde{\mathbf{W}} \in \mathbb{R}^{d \times m}$. *Consider Assumption 1, Assumption 2, and assume that* $\mathcal{L}_S(\widetilde{\mathbf{W}}) \leq C_0^2$. *Then, for any* $S$,

$$\lambda_{\max}(\nabla^2 \mathcal{L}_S(\mathbf{W})) \leq \rho \quad \text{where} \quad \rho \stackrel{\text{def}}{=} C_x^2 \left( B_{\phi'}^2 + B_{\phi''} B_\phi + \frac{B_{\phi''} C_y}{\sqrt{m}} \right),$$

$$\min_{\alpha \in [0,1]} \lambda_{\min}(\nabla^2 \mathcal{L}_S(\widetilde{\mathbf{W}} + \alpha(\mathbf{W} - \widetilde{\mathbf{W}}))) \geq -\frac{B_{\phi''}(B_{\phi'} C_x + C_0)}{\sqrt{m}} \cdot (1 \vee \|\mathbf{W} - \widetilde{\mathbf{W}}\|_F). \quad (10)$$

*Proof.* Vectorising allows the loss's Hessian to be denoted

$$\nabla^2 \ell(\mathbf{W}, z) = \nabla f_{\mathbf{W}}(\mathbf{x}) \nabla f_{\mathbf{W}}(\mathbf{x})^\top + \nabla^2 f_{\mathbf{W}}(\mathbf{x})(f_{\mathbf{W}}(\mathbf{x}) - y) \quad (11)$$

where

$$\nabla f_{\mathbf{W}}(\mathbf{x}) = \begin{pmatrix} u_1 \mathbf{x} \phi'(\langle (\mathbf{W})_1, \mathbf{x} \rangle) \\ u_2 \mathbf{x} \phi'(\langle (\mathbf{W})_2, \mathbf{x} \rangle) \\ \vdots \\ u_m \mathbf{x} \phi'(\langle (\mathbf{W})_m, \mathbf{x} \rangle) \end{pmatrix} \in \mathbb{R}^{dm}$$

and $\nabla^2 f_{\mathbf{W}}(\mathbf{x}) \in \mathbb{R}^{dm \times dm}$ with

$$\nabla^2 f_{\mathbf{W}}(\mathbf{x}) = \begin{pmatrix} u_1 \mathbf{x} \mathbf{x}^\top \phi''(\langle (\mathbf{W})_1, \mathbf{x} \rangle) & 0 & 0 & \ldots & 0 \\ 0 & u_2 \mathbf{x} \mathbf{x}^\top \phi''(\langle (\mathbf{W})_2, \mathbf{x} \rangle) & 0 & \ldots & 0 \\ \vdots & & \ddots & \ddots & \vdots & \vdots \\ 0 & 0 & 0 & \ldots & u_m \mathbf{x} \mathbf{x}^\top \phi''(\langle (\mathbf{W})_m, \mathbf{x} \rangle) \end{pmatrix}$$

Note that we then immediately have with $\mathbf{v} = (\mathbf{v}_1, \mathbf{v}_2, \ldots, \mathbf{v}_m) \in \mathbb{R}^{dm}$ with $\mathbf{v}_i \in \mathbb{R}^d$

$$\|\nabla^2 f_{\mathbf{W}}(\mathbf{x})\|_2 = \max_{\mathbf{v}:\|\mathbf{v}\|_2 \leq 1} \sum_{j=1}^m u_j \langle \mathbf{v}_j, \mathbf{x} \rangle^2 \phi''(\langle (\mathbf{W})_j, \mathbf{x} \rangle)$$

$$\leq \frac{1}{\sqrt{m}} \|\mathbf{x}\|_2^2 B_{\phi''} \max_{\mathbf{v}:\|\mathbf{v}\|_2 \leq 1} \sum_{j=1}^m \|\mathbf{v}_j\|_2^2$$

$$\leq \frac{C_x^2 B_{\phi''}}{\sqrt{m}}. \quad (12)$$

We then see that the maximum Eigenvalue of the Hessian is upper bounded for any $\mathbf{W} \in \mathbb{R}^{dm}$, that is

$$\|\nabla^2 \ell(\mathbf{W}, z)\|_2 \leq \|\nabla f_{\mathbf{W}}(\mathbf{x})\|_2^2 + \|\nabla^2 f_{\mathbf{W}}(\mathbf{x})\|_2 |f_{\mathbf{W}}(\mathbf{x}) - y| \quad (13)$$

$$\leq C_x^2 B_{\phi'}^2 + \frac{C_x^2 B_{\phi''}}{\sqrt{m}}(\sqrt{m} B_\phi + C_y) \quad (14)$$

and therefore the objective is $\rho$-smooth with $\rho = C_x^2\big(B_{\phi'}^2 + B_{\phi''}B_\phi + \frac{B_{\phi''}C_y}{\sqrt{m}}\big)$.

Let us now prove the lower bound (10). For some fixed $\mathbf{W}, \widetilde{\mathbf{W}} \in \mathbb{R}^{d\times m}$ define

$$\mathbf{W}(\alpha) \stackrel{\text{def}}{=} \widetilde{\mathbf{W}} + \alpha(\mathbf{W} - \widetilde{\mathbf{W}}) \qquad \alpha \in [0,1] \ .$$

Looking at the Hessian in (11), the first matrix is positive semi-definite, therefore

$$\lambda_{\min}(\nabla^2 \mathcal{L}_S(\mathbf{W}(\alpha))) \geq -\Big(\max_{i=1,\ldots,n}\big\{\|\nabla^2 f_{\mathbf{W}(\alpha)}(\mathbf{x}_i)\|_2\big\}\Big)\frac{1}{n}\sum_{i=1}^n |f_{\mathbf{W}(\alpha)}(\mathbf{x}_i) - y_i|$$

$$\geq -\frac{C_x^2 B_{\phi''}}{\sqrt{m}} \cdot \frac{1}{n}\sum_{i=1}^n |f_{\mathbf{W}(\alpha)}(\mathbf{x}_i) - y_i|$$

where we have used the upper bound on $\|\nabla^2 f_{\mathbf{W}}(\mathbf{x}_i)\|_2$. Adding and subtracting $f_{\widetilde{\mathbf{W}}}(\mathbf{x}_i)$ inside the absolute value we then get

$$\frac{1}{n}\sum_{i=1}^n |f_{\mathbf{W}(\alpha)}(\mathbf{x}_i) - y_i| \leq \frac{1}{n}\sum_{i=1}^n |f_{\mathbf{W}(\alpha)}(\mathbf{x}_i) - f_{\widetilde{\mathbf{W}}}(\mathbf{x}_i)| + \frac{1}{n}\sum_{i=1}^n |f_{\widetilde{\mathbf{W}}}(\mathbf{x}_i) - y_i|$$

$$\leq B_{\phi'}C_x\|\mathbf{W}(\alpha) - \widetilde{\mathbf{W}}\|_2 + \sqrt{\mathcal{L}_S(\widetilde{\mathbf{W}})}$$

$$\leq B_{\phi'}C_x\|\mathbf{W}(\alpha) - \widetilde{\mathbf{W}}\|_2 + \sqrt{\mathcal{L}_S(\mathbf{W}_0)}$$

$$\leq \big(B_{\phi'}C_x + C_0\big)(1 \vee \|\mathbf{W}(\alpha) - \widetilde{\mathbf{W}}\|_2)$$

where for the second term we have simply applied Cauchy-Schwarz inequality. For the first term, we used that for any $\mathbf{W}, \widetilde{\mathbf{W}} \in \mathbb{R}^{dm}$ we see that

$$|f_{\mathbf{W}}(\mathbf{x}) - f_{\widetilde{\mathbf{W}}}(\mathbf{x})| \leq \frac{1}{\sqrt{m}}\sum_{i=1}^m |\phi(\langle (\mathbf{W})_i, \mathbf{x}\rangle) - \phi(\langle (\widetilde{\mathbf{W}})_i, \mathbf{x}\rangle)| \tag{15}$$

$$\leq \frac{B_{\phi'}}{\sqrt{m}}\sum_{i=1}^m \Big|\langle (\mathbf{W})_i - (\widetilde{\mathbf{W}})_i, \mathbf{x}\rangle\Big|$$

$$\leq C_x B_{\phi'}\|\mathbf{W} - \widetilde{\mathbf{W}}\|_2. \tag{16}$$

Bringing everything together yields the desired lower bound

$$\lambda_{\min}(\nabla^2 \mathcal{L}_S(\mathbf{W}(\alpha))) \geq -\frac{C_x^2}{\sqrt{m}}B_{\phi''}\big(B_{\phi'}C_x + C_0\big)(1 \vee \|\mathbf{W}(\alpha) - \widetilde{\mathbf{W}}\|_2)$$

$$\geq -\frac{C_x^2}{\sqrt{m}}B_{\phi''}\big(B_{\phi'}C_x + C_0\big)(1 \vee \|\mathbf{W} - \widetilde{\mathbf{W}}\|_2) \ .$$

This holds for any $\alpha \in [0,1]$, therefore, we took the minimum. $\qquad\square$

## B  Optimisation Error Bound (Proof of Lemma 2)

In this section we present the proof for the Optimisation Error term. We begin by quoting the result which we set to prove.

**Lemma 2 (restated).** *Consider Assumptions 1 and 2. Fix $t > 0$. If $\eta \leq 1/(2\rho)$, then*

$$\frac{1}{t}\sum_{j=0}^t \mathcal{L}_S(\mathbf{W}_j) \leq \min_{\mathbf{W}\in\mathbb{R}^{d\times m}}\left\{\mathcal{L}_S(\mathbf{W}) + \frac{\|\mathbf{W} - \mathbf{W}_0\|_F^2}{\eta t} + \frac{\widetilde{b}\|\mathbf{W} - \mathbf{W}_0\|_F^3}{\sqrt{m}}\right\} + \widetilde{b}C_0 \cdot \frac{(\eta t)^{\frac{3}{2}}}{\sqrt{m}}$$

*where $\widetilde{b} = C_x^2 B_{\phi''}(B_{\phi'}C_x + C_0)$.*

*Proof.* Using Lemma 1 as well as that $\eta\rho \leq 1$ from the assumption within the theorem yields for $t \geq 0$

$$\mathcal{L}_S(\mathbf{W}_{t+1}) \leq \mathcal{L}_S(\mathbf{W}_t) - \eta\big(1 - \frac{\eta\rho}{2}\big)\|\nabla\mathcal{L}_S(\mathbf{W}_t)\|_2^2$$

$$\leq \mathcal{L}_S(\mathbf{W}_t) - \frac{\eta}{2}\|\nabla\mathcal{L}_S(\mathbf{W}_t)\|_2^2.$$

Fix some $\mathbf{W} \in \mathbb{R}^{dm}$. We then use the following inequality which will be proven shortly:

$$\mathcal{L}_S(\mathbf{W}_t) \leq \mathcal{L}_S(\mathbf{W}) - \langle \mathbf{W} - \mathbf{W}_t, \nabla \mathcal{L}_S(\mathbf{W}_t) \rangle + \frac{\widetilde{b}}{\sqrt{m}} \left(1 \vee \|\mathbf{W} - \mathbf{W}_t\|_2\right)^3 \qquad (17)$$

Plugging in this inequality we then get

$$\mathcal{L}_S(\mathbf{W}_{t+1}) \leq \mathcal{L}_S(\mathbf{W}) - \langle \mathbf{W} - \mathbf{W}_t, \nabla \mathcal{L}_S(\mathbf{W}_t) \rangle - \frac{\eta}{2}\|\nabla \mathcal{L}_S(\mathbf{W}_t)\|_2^2 + \frac{\widetilde{b}}{\sqrt{m}}\left(1 \vee \|\mathbf{W} - \mathbf{W}_t\|_2\right)^3.$$

Note that we can rewrite

$$- \langle \mathbf{W} - \mathbf{W}_t, \nabla \mathcal{L}_S(\mathbf{W}_t) \rangle - \frac{\eta}{2}\|\nabla \mathcal{L}_S(\mathbf{W}_t)\|_2^2$$
$$= \frac{1}{\eta}\langle \mathbf{W} - \mathbf{W}_t, \mathbf{W}_{t+1} - \mathbf{W}_t \rangle - \frac{1}{2\eta}\|\mathbf{W}_{t+1} - \mathbf{W}_t\|_2^2$$
$$= \frac{1}{\eta}\left(\|\mathbf{W} - \mathbf{W}_t\|_2^2 - \|\mathbf{W}_{t+1} - \mathbf{W}\|_2^2\right)$$

where we used that for any vectors $\mathbf{x}, \mathbf{y}, \mathbf{z}$: $2\langle \mathbf{x} - \mathbf{y}, \mathbf{x} - \mathbf{z} \rangle = \|\mathbf{x} - \mathbf{y}\|_2^2 + \|\mathbf{x} - \mathbf{z}\|_2^2 - \|\mathbf{y} - \mathbf{z}\|_2^2$ (which is easier to see if we relabel $2\langle \mathbf{a}, \mathbf{b} \rangle = \|\mathbf{a}\|_2^2 + \|\mathbf{b}\|_2^2 - \|\mathbf{a} - \mathbf{b}\|_2^2$). Plugging in and summing up we get

$$\frac{1}{t}\sum_{s=0}^{t}\mathcal{L}_S(\mathbf{W}_t) \leq \mathcal{L}_S(\mathbf{W}) + \frac{\|\mathbf{W} - \mathbf{W}_0\|_2^2}{\eta t} + \frac{\widetilde{b}}{\sqrt{m}} \cdot \frac{1}{t}\sum_{s=0}^{t}\left(1 \vee \|\mathbf{W} - \mathbf{W}_t\|_2\right)^3.$$

Since the choice of $\mathbf{W}$ was arbitrary, we can simply take the minimum.

**Proof of Eq. (17).** Let us now prove the key Eq. (17). Fix $t \geq 0$, and let us define the following functions for $\alpha \in [0, 1]$

$$\mathbf{W}(\alpha) \stackrel{\text{def}}{=} \mathbf{W}_t + \alpha(\mathbf{W} - \mathbf{W}_t),$$
$$g(\alpha) \stackrel{\text{def}}{=} \mathcal{L}_S(\mathbf{W}(\alpha)) + \frac{\widetilde{b}}{\sqrt{m}} \cdot \frac{\alpha^2}{2}\left(1 \vee \|\mathbf{W} - \mathbf{W}_t\|_2\right)^3.$$

Note that computing the derivative we have

$$g''(\alpha) = (\mathbf{W} - \mathbf{W}_t)^{\top}\nabla^2\mathcal{L}_S(\mathbf{W}(\alpha))(\mathbf{W} - \mathbf{W}_t) + \frac{\widetilde{b}}{\sqrt{m}}\left(1 \vee \|\mathbf{W} - \mathbf{W}_t\|_2\right)^3.$$

On the other hand by Lemma 1 we have

$$\min_{\alpha \in [0,1]} \lambda_{\min}(\nabla^2\mathcal{L}_S(\mathbf{W}(\alpha))) \geq -\frac{\widetilde{b}}{\sqrt{m}}\left(1 \vee \|\mathbf{W} - \mathbf{W}_t\|_2\right)$$

and we immediately have $g''(\alpha) \geq 0$, and thus, $g(\cdot)$ is convex on $[0, 1]$. Inequality (17) then arises from $g(1) - g(0) \geq g'(0)$, in particular

$$g(1) - g(0) = \mathcal{L}_S(\mathbf{W}) + \frac{\widetilde{b}}{\sqrt{m}}\left(1 \vee \|\mathbf{W} - \mathbf{W}_t\|_2\right)^3 - \mathcal{L}_S(\mathbf{W}_t)$$
$$\geq \langle \mathbf{W} - \mathbf{W}_t, \nabla \mathcal{L}_S(\mathbf{W}_t) \rangle$$
$$= g'(0)$$

as required. $\qquad \square$

## C  Generalisation Gap Bound (Proof of Theorem 1)

In this section we prove:

**Theorem 1 (restated).** *Consider Assumptions 1 and 2. Fix $t > 0$. If $\eta \leq 1/(2\rho)$ and*

$$m \geq 144(\eta t)^2 C_x^4 C_0^2 B_{\phi''}^2 \left(4B_{\phi'} C_x \sqrt{\eta t} + \sqrt{2}\right)^2$$

then $\quad \mathbf{E}\left[\epsilon^{Gen}(\mathbf{W}_{t+1}) \mid \mathbf{W}_0, \mathbf{u}\right] \leq b\left(\frac{\eta}{n} + \frac{\eta^2 t}{n^2}\right) \sum_{j=0}^{t} \mathbf{E}\left[\mathcal{L}_S(\mathbf{W}_j) \mid \mathbf{W}_0, \mathbf{u}\right]$

*where $b = 16e^3 C_x^{\frac{3}{2}} B_{\phi'}^2 (1 + C_x^{\frac{3}{2}} B_{\phi'}^2)$.*

To prove this result we use algorithmic stability arguments. Recall that we can write [Shalev-Shwartz and Ben-David, 2014, Chapter 13],

$$\mathbf{E}\left[\mathcal{L}(\mathbf{W}_{t+1}) - \mathcal{L}_S(\mathbf{W}_{t+1}) \mid \mathbf{W}_0, \mathbf{u}\right] = \frac{1}{n} \sum_{i=1}^{n} \mathbf{E}\left[\ell(\mathbf{W}_{t+1}, \widetilde{z}_i) - \ell(\mathbf{W}_{t+1}^{(i)}, \widetilde{z}_i) \mid \mathbf{W}_0, \mathbf{u}\right] .$$

The following lemma shown in Appendix C.1 then bounds the Generalisation error in terms of a notation of stability.

**Lemma 3 (restated).** *Consider Assumptions 1 and 2. Then, for any $t \geq 0$,*

$$\mathbf{E}\left[\mathcal{L}(\mathbf{W}_t) - \mathcal{L}_S(\mathbf{W}_t) \mid \mathbf{W}_0, \mathbf{u}\right]$$

$$\leq B_{\phi'} \sqrt{C_x} \sqrt{\mathbf{E}\left[\mathcal{L}_S(\mathbf{W}_t) \mid \mathbf{W}_0, \mathbf{u}\right]} \sqrt{\frac{1}{n} \sum_{i=1}^{n} \mathbf{E}\left[\|\mathbf{W}_t - \mathbf{W}_t^{(i)}\|_{op}^2 \mid \mathbf{W}_0, \mathbf{u}\right]}$$

$$+ C_x B_{\phi'}^2 \cdot \frac{1}{n} \sum_{i=1}^{n} \mathbf{E}\left[\|\mathbf{W}_t - \mathbf{W}_t^{(i)}\|_{op}^2 \mid \mathbf{W}_0, \mathbf{u}\right]$$

*where $\| \cdot \|_{op}$ denotes the spectral norm.*

We note while the stability is only required on the spectral norm, our bound will be on the element wise $L_2$-norm i.e. Frobenius norm, which upper bounds the spectral norm. It is summarised within the following lemma shown in Appendix C.2.

**Lemma 4** (Bound on On-Average Parameter Stability). *Consider Assumptions 1 and 2. Fix $t > 0$. If $\eta \leq 1/(2\rho)$, then*

$$\frac{1}{n} \sum_{i=1}^{n} \mathbf{E}\left[\|\mathbf{W}_{t+1} - \mathbf{W}_{t+1}^{(i)}\|_F^2 \mid \mathbf{W}_0, \mathbf{u}\right] \leq 8e \frac{\eta^2 t}{n^2} \left(\frac{1}{1 - 2\eta\epsilon}\right)^t \frac{1}{n} \sum_{i=1}^{n} \sum_{j=0}^{t} \mathbf{E}\left[\|\nabla\ell(\mathbf{W}_j, z_i)\|_2^2 \mid \mathbf{W}_0, \mathbf{u}\right]$$

*where $\epsilon = 2 \cdot \frac{C_x^2 \sqrt{C_0} B_{\phi''}}{\sqrt{m}} \left(4B_{\phi'} C_x \sqrt{\eta t} + \sqrt{2}\right)$.*

Theorem 1 then arises by combining Lemma 3 and Lemma 4, and noting the following three points. Firstly, recall that

$$\frac{1}{n} \sum_{i=1}^{n} \|\nabla\ell(\mathbf{W}, z_i)\|_2^2 \leq \left(\max_{i=1,\ldots,n} \|\nabla f_{\mathbf{W}}(\mathbf{x}_i)\|_2^2\right) \frac{1}{n} \sum_{i=1}^{n} (f_{\mathbf{W}}(\mathbf{x}_i) - y_i)^2$$

$$\leq 2C_x^2 B_{\phi'}^2 \mathcal{L}_S(\mathbf{W}).$$

Secondly, note that we have $\left(\frac{1}{1-2\eta\epsilon}\right)^t \leq \exp\left(\frac{2\eta t\epsilon}{1-2\eta t\epsilon}\right) \leq e^2$ when $2\eta t\epsilon \leq 2/3$. For this to occur we then require

$$\epsilon = 2 \cdot \frac{C_x^2 \sqrt{C_0} B_{\phi''}}{\sqrt{m}} \cdot \left(4B_{\phi'} C_x \sqrt{\eta t} + \sqrt{2}\right) \leq \frac{1}{3\eta t},$$

which is satisfied by scaling $m$ sufficient large, in particular, as required within condition (7) within the statement of Theorem 1. This allows us to arrive at the bound on the $L_2$-stability

$$\frac{1}{n} \sum_{i=1}^{n} \mathbf{E}\left[\|\mathbf{W}_{t+1} - \mathbf{W}_{t+1}^{(i)}\|_F^2 \mid \mathbf{W}_0, \mathbf{u}\right] \leq \frac{\eta^2 t}{n^2} \cdot 16e^3 C_x^2 B_{\phi'}^2 \sum_{j=0}^{t} \mathbf{E}\left[\mathcal{L}_S(\mathbf{W}_j) \mid \mathbf{W}_0, \mathbf{u}\right] .$$

Third and finally, note that we can bound

$$\sqrt{\mathbf{E}\left[\mathcal{L}_S(\mathbf{W}_{t+1}) \mid \mathbf{W}_0, \mathbf{u}\right]}\sqrt{\frac{\eta^2 t}{n^2}\sum_{j=0}^{t}\mathbf{E}\left[\mathcal{L}_S(\mathbf{W}_j) \mid \mathbf{W}_0, \mathbf{u}\right]}$$

$$= \frac{\eta}{n}\sqrt{t\mathbf{E}\left[\mathcal{L}_S(\mathbf{W}_{t+1}) \mid \mathbf{W}_0, \mathbf{u}\right]}\sqrt{\sum_{j=0}^{t}\mathbf{E}\left[\mathcal{L}_S(\mathbf{W}_j) \mid \mathbf{W}_0, \mathbf{u}\right]}$$

$$\leq \frac{\eta}{n}\sum_{j=0}^{t}\mathbf{E}\left[\mathcal{L}_S(\mathbf{W}_j) \mid \mathbf{W}_0, \mathbf{u}\right]$$

since $\mathcal{L}_S(\mathbf{W}_{t+1}) \leq \frac{1}{t}\sum_{j=1}^{t}\mathcal{L}_S(\mathbf{W}_j)$. This then results in

$$\mathbf{E}\left[\mathcal{L}(\mathbf{W}_{t+1}) - \mathcal{L}_S(\mathbf{W}_{t+1}) \mid \mathbf{W}_0, \mathbf{u}\right]$$

$$\leq \left(\frac{\eta}{n}\left(4e^2 C_x^{3/2}B_{\phi'}^2\right) + \frac{\eta^2 t}{n^2}\left(16e^3 C_x^3 B_{\phi'}^4\right)\right)\sum_{j=0}^{t}\mathbf{E}\left[\mathcal{L}_S(\mathbf{W}_j) \mid \mathbf{W}_0, \mathbf{u}\right]$$

$$\leq 16e^3 C_x^{3/2}B_{\phi'}^2(1 + C_x^{3/2}B_{\phi'}^2)\left(\frac{\eta}{n} + \frac{\eta^2 t}{n^2}\right)\sum_{j=0}^{t}\mathbf{E}\left[\mathcal{L}_S(\mathbf{W}_j) \mid \mathbf{W}_0, \mathbf{u}\right]$$

as required.

### C.1  Proof of Lemma 3: From loss stability to parameter stability

Recall that $\widetilde{z}_i = (\widetilde{\mathbf{x}}_i, y_i) \in \mathcal{B}_2^d(C_x) \times [-C_y, C_y]$. Expanding the square loss and some basic algebra gives us:

$$2\left(\ell(\mathbf{W}_t, \widetilde{z}_i) - \ell(\mathbf{W}_t^{(i)}, \widetilde{z}_i)\right)$$

$$= (f_{\mathbf{W}_t}(\widetilde{\mathbf{x}}_i) - \widetilde{y}_i)^2 - \left(f_{\mathbf{W}_t^{(i)}}(\widetilde{\mathbf{x}}_i) - \widetilde{y}_i\right)^2$$

$$= (f_{\mathbf{W}_t}(\widetilde{\mathbf{x}}_i) - \widetilde{y}_i)\left(f_{\mathbf{W}_t}(\widetilde{\mathbf{x}}_i) - f_{\mathbf{W}_t^{(i)}}(\widetilde{\mathbf{x}}_i)\right) + \left(f_{\mathbf{W}_t^{(i)}}(\widetilde{\mathbf{x}}_i) - \widetilde{y}_i\right)\left(f_{\mathbf{W}_t}(\widetilde{\mathbf{x}}_i) - f_{\mathbf{W}_t^{(i)}}(\widetilde{\mathbf{x}}_i)\right)$$

$$= \left(f_{\mathbf{W}_t}(\widetilde{\mathbf{x}}_i) - f_{\mathbf{W}_t^{(i)}}(\widetilde{\mathbf{x}}_i)\right)^2 + 2\left(f_{\mathbf{W}_t^{(i)}}(\widetilde{\mathbf{x}}_i) - \widetilde{y}_i\right)\left(f_{\mathbf{W}_t}(\widetilde{\mathbf{x}}_i) - f_{\mathbf{W}_t^{(i)}}(\widetilde{\mathbf{x}}_i)\right) \ .$$

We then have

$$\frac{1}{n}\sum_{i=1}^{n}\mathbf{E}\left[\ell(\mathbf{W}_t, \widetilde{z}_i) - \ell(\mathbf{W}_t^{(i)}, \widetilde{z}_i) \ \middle| \ \mathbf{W}_0, \mathbf{u}\right]$$

$$\leq \frac{1}{n}\sum_{i=1}^{n}\mathbf{E}\left[\left|\left(f_{\mathbf{W}_t^{(i)}}(\widetilde{\mathbf{x}}_i) - \widetilde{y}_i\right)\right|\left|\left(f_{\mathbf{W}_t}(\widetilde{\mathbf{x}}_i) - f_{\mathbf{W}_t^{(i)}}(\widetilde{\mathbf{x}}_i)\right)\right| \ \middle| \ \mathbf{W}_0, \mathbf{u}\right]$$

$$+ \frac{1}{2n}\sum_{i=1}^{n}\mathbf{E}\left[\left(f_{\mathbf{W}_t}(\widetilde{\mathbf{x}}_i) - f_{\mathbf{W}_t^{(i)}}(\widetilde{\mathbf{x}}_i)\right)^2 \ \middle| \ \mathbf{W}_0, \mathbf{u}\right]$$

$$\leq \sqrt{\frac{1}{n}\sum_{i=1}^{n}\mathbf{E}\left[\left(f_{\mathbf{W}_t^{(i)}}(\widetilde{\mathbf{x}}_i) - \widetilde{y}_i\right)^2 \ \middle| \ \mathbf{W}_0, \mathbf{u}\right]}\sqrt{\frac{1}{n}\sum_{i=1}^{n}\mathbf{E}\left[\left(f_{\mathbf{W}_t}(\widetilde{\mathbf{x}}_i) - f_{\mathbf{W}_t^{(i)}}(\widetilde{\mathbf{x}}_i)\right)^2 \ \middle| \ \mathbf{W}_0, \mathbf{u}\right]}$$

$$+ \frac{1}{2n}\sum_{i=1}^{n}\mathbf{E}\left[\left(f_{\mathbf{W}_t}(\widetilde{\mathbf{x}}_i) - f_{\mathbf{W}_t^{(i)}}(\widetilde{\mathbf{x}}_i)\right)^2 \ \middle| \ \mathbf{W}_0, \mathbf{u}\right]$$

where performing steps as in Eq. (15)-(16) we have

$$\left(f_{\mathbf{W}_t}(\widetilde{\mathbf{x}}_i) - f_{\mathbf{W}_t^{(i)}}(\widetilde{\mathbf{x}}_i)\right)^2 \leq C_x^2 B_{\phi'}^2 \|\mathbf{W}_t - \mathbf{W}_t^{(i)}\|_2^2 \ .$$

Plugging in this bound then yields the result.

## C.2  Proof of Lemma 4: Bound on on-average parameter stability

Throughout the proof empirical risk w.r.t. remove-one tuple $S^{\backslash i}$ is denoted as

$$\mathcal{L}_{S^{\backslash i}}(\mathbf{W}) = \mathcal{L}_S(\mathbf{W}) - \frac{1}{n}\ell(\mathbf{W}, z_i) = \mathcal{L}_{S_i}(\mathbf{W}) - \frac{1}{n}\ell(\mathbf{W}, \widetilde{z}_i) .$$

Plugging in the gradient updates with the inequality $(a+b)^2 \leq (1+p)a^2 + (1+1/p)b^2$ for $p > 0$ then yields (this technique having been applied within [Lei and Ying, 2020])

$$\|\mathbf{W}_{t+1} - \mathbf{W}_{t+1}^{(i)}\|_2^2 \leq (1+p) \underbrace{\|\mathbf{W}_t - \mathbf{W}_t^{(i)} - \eta\left(\nabla\mathcal{L}_{S^{\backslash i}}(\mathbf{W}_t) - \nabla\mathcal{L}_{S^{\backslash i}}(\mathbf{W}_t^{(i)})\right)\|_2^2}_{\text{Expansiveness of the Gradient Update}}$$

$$+ (1+1/p) \cdot \frac{2\eta^2}{n^2} \cdot \left(\|\nabla\ell(\mathbf{W}_t, z_i)\|_2^2 + \|\nabla\ell(\mathbf{W}_t^{(i)}), \widetilde{z}_i)\|_2^2\right) .$$

We must now bound the expansiveness of the gradient update. Opening-up the squared norm we get

$$\|\mathbf{W}_t - \mathbf{W}_t^{(i)} - \eta(\nabla\mathcal{L}_{S^{\backslash i}}(\mathbf{W}_t) - \nabla\mathcal{L}_{S^{\backslash i}}(\mathbf{W}_t^{(i)}))\|_2^2$$
$$= \|\mathbf{W}_t - \mathbf{W}_t^{(i)}\|_2^2 + \eta^2\|\nabla\mathcal{L}_{S^{\backslash i}}(\mathbf{W}_t) - \nabla\mathcal{L}_{S^{\backslash i}}(\mathbf{W}_t^{(i)})\|_2^2$$
$$- 2\eta\left\langle\mathbf{W}_t - \mathbf{W}_t^{(i)}, \nabla\mathcal{L}_{S^{\backslash i}}(\mathbf{W}_t) - \nabla\mathcal{L}_{S^{\backslash i}}(\mathbf{W}_t^{(i)})\right\rangle$$

For this purpose we will use the following key lemma shown in Appendix C.3.1:

**Lemma 5** (Almost Co-coercivity of the Gradient Operator). *Consider the assumptions of Lemma 4. Then for $t \geq 1$*

$$\left\langle\mathbf{W}_t - \mathbf{W}_t^{(i)}, \nabla\mathcal{L}_{S^{\backslash i}}(\mathbf{W}_t) - \nabla\mathcal{L}_{S^{\backslash i}}(\mathbf{W}_t^{(i)})\right\rangle \geq 2\eta\left(1 - \frac{\eta\rho}{2}\right)\|\nabla\mathcal{L}_{S^{\backslash i}}(\mathbf{W}_t) - \nabla\mathcal{L}_{S^{\backslash i}}(\mathbf{W}_t^{(i)})\|_2^2$$

$$- \epsilon\left\|\mathbf{W}_t - \mathbf{W}_t^{(i)} - \eta\left(\nabla\mathcal{L}_{S^{\backslash i}}(\mathbf{W}_t) - \nabla\mathcal{L}_{S^{\backslash i}}(\mathbf{W}_t^{(i)})\right)\right\|_2^2$$

*where*

$$\rho = C_x^2\left(B_{\phi'}^2 + B_{\phi''}B_\phi + \frac{B_{\phi''}C_y}{\sqrt{m}}\right) ,$$

$$\epsilon = 2 \cdot \frac{C_x^2\sqrt{C_0}B_{\phi''}}{\sqrt{m}}\left(4B_{\phi'}C_x\sqrt{\eta t} + \sqrt{2}\right) .$$

Thus by Lemma 5 we get

$$\|\mathbf{W}_t - \mathbf{W}_t^{(i)} - \eta\left(\nabla\mathcal{L}_{S^{\backslash i}}(\mathbf{W}_t) - \nabla\mathcal{L}_{S^{\backslash i}}(\mathbf{W}_t^{(i)})\right)\|_2^2$$

$$\leq \|\mathbf{W}_t - \mathbf{W}_t^{(i)}\|_2^2 + \eta^2(2\eta\rho - 3)\left\|\nabla\mathcal{L}_{S^{\backslash i}}(\mathbf{W}_t) - \nabla\mathcal{L}_{S^{\backslash i}}(\mathbf{W}_t^{(i)})\right\|_2^2$$

$$+ 2\eta\epsilon\left\|\mathbf{W}_t - \mathbf{W}_t^{(i)} - \eta\left(\nabla\mathcal{L}_{S^{\backslash i}}(\mathbf{W}_t) - \nabla\mathcal{L}_{S^{\backslash i}}(\mathbf{W}_t^{(i)})\right)\right\|_2^2 .$$

Rearranging and using that $\eta\rho \leq 1/2$ we then arrive at the recursion

$$\|\mathbf{W}_{t+1} - \mathbf{W}_{t+1}^{(i)}\|_F^2 \leq \frac{1+p}{1-2\eta\epsilon} \cdot \|\mathbf{W}_t - \mathbf{W}_t^{(i)}\|_F^2$$

$$+ \left(1 + \frac{1}{p}\right) \cdot \frac{2\eta^2}{n^2}\left(\|\nabla\ell(\mathbf{W}_t, z_i)\|_2^2 + \|\nabla\ell(\mathbf{W}_t^{(i)}), \widetilde{z}_i)\|_2^2\right)$$

$$\leq \left(1 + \frac{1}{p}\right) \cdot \frac{2\eta^2}{n^2}\left(\frac{1+p}{1-2\eta\epsilon}\right)^t \sum_{j=0}^t\left(\|\nabla\ell(\mathbf{W}_j, z_i)\|_2^2 + \|\nabla\ell(\mathbf{W}_j^{(i)}), \widetilde{z}_i)\|_2^2\right) .$$

Taking expectation and summing we then get

$$\frac{1}{n}\sum_{i=1}^n \mathbf{E}\left[\|\mathbf{W}_{t+1} - \mathbf{W}_{t+1}^{(i)}\|_F^2 \mid \mathbf{W}_0, \mathbf{u}\right]$$

$$\leq 4(1+1/p)\frac{2\eta^2}{n^2}\left(\frac{(1+p)}{1-2\eta\epsilon}\right)^t \sum_{j=0}^t \mathbf{E}\left[\|\nabla\ell(\mathbf{W}_j, z_i)\|_2^2 \mid \mathbf{W}_0, \mathbf{u}\right]$$

where we note that $\mathbf{E}\left[\|\nabla\ell(\mathbf{W}_j, z_i)\|_2^2 \mid \mathbf{W}_0, \mathbf{u}\right] = \mathbf{E}\left[\|\nabla\ell(\mathbf{W}_j^{(i)}, \widetilde{z}_i)\|_2^2 \mid \mathbf{W}_0, \mathbf{u}\right]$ since $z_i$ and $\widetilde{z}_i$ are identically distributed. Picking $p = 1/t$ and noting that $(1+p)^t = (1+1/t)^t \leq e$ yields the bound.

### C.3 Proof of Lemma 5: Almost-co-coercivity of the Gradient Operator

In this section we show Lemma 5 which says that a gradient operator of an overparameterised shallow network is almost-co-coercive. The proof of this lemma will require two auxiliary lemmas.

**Lemma 6.** *Consider Assumptions 1 and 2 and assume that $\eta \leq 1/(2\rho)$. Then for any $t \geq 0$, $i \in [n]$,*

$$\|\mathbf{W}_t - \mathbf{W}_0\|_F \leq \sqrt{2\eta t \mathcal{L}_S(\mathbf{W}_0)}\,,$$
$$\|\mathbf{W}_t^{(i)} - \mathbf{W}_0\|_F \leq \sqrt{2\eta t \mathcal{L}_{S^{(i)}}(\mathbf{W}_0)}\,.$$

*Proof.* The proof is given in Appendix C.3.2. $\qquad\square$

We also need the following Lemma (whose proof is very similar to Lemma 1).

**Lemma 7.** *Consider Assumptions 1 and 2. Fix $s \geq 0$, $i \in [n]$. For any $\alpha \in [0,1]$ denote*

$$\mathbf{W}(\alpha) \stackrel{\text{def}}{=} \mathbf{W}_s^{(i)} + \alpha\left(\mathbf{W}_s - \mathbf{W}_s^{(i)} - \eta\left(\nabla\mathcal{L}_{S\backslash i}(\mathbf{W}_s) - \nabla\mathcal{L}_{S\backslash i}(\mathbf{W}_s^{(i)})\right)\right)\,,$$
$$\widetilde{\mathbf{W}}(\alpha) \stackrel{\text{def}}{=} \mathbf{W}_s + \alpha\left(\mathbf{W}_s^{(i)} - \mathbf{W}_s - \eta\left(\nabla\mathcal{L}_{S\backslash i}(\mathbf{W}_s^{(i)}) - \nabla\mathcal{L}_{S\backslash i}(\mathbf{W}_s)\right)\right)\,.$$

*If $\eta \leq 1/(2\rho)$, then*

$$\min_{\alpha\in[0,1]} \lambda_{\min}\left(\nabla^2\mathcal{L}_{S\backslash i}\left(\mathbf{W}(\alpha)\right)\right) \geq -\widetilde{\epsilon}\,,$$
$$\min_{\alpha\in[0,1]} \lambda_{\min}\left(\nabla^2\mathcal{L}_{S\backslash i}\left(\widetilde{\mathbf{W}}(\alpha)\right)\right) \geq -\widetilde{\epsilon}\,.$$

*with*

$$\widetilde{\epsilon} = \frac{C_x^2 B_{\phi''}}{\sqrt{m}}\left(4B_{\phi'}C\sqrt{\eta s}\left(\sqrt{\mathcal{L}_S(\mathbf{W}_0)} + \sqrt{\mathcal{L}_{S_i}(\mathbf{W}_0)}\right) + \sqrt{2\mathcal{L}_{S_i}(\mathbf{W}_0)} + \sqrt{2\mathcal{L}_S(\mathbf{W}_0)}\right)\,.$$

*Proof.* The proof is given in Appendix C.3.3. $\qquad\square$

#### C.3.1 Proof of Lemma 5

The proof of this Lemma follows by arguing that the operator $\mathbf{w} \mapsto \nabla\mathcal{L}_{S\backslash i}(\mathbf{w})$ is almost-co-coercive: Recall that the operator $F : \mathcal{X} \to \mathcal{X}$ is co-coercive whenever $\langle \nabla F(\mathbf{x}) - \nabla F(\mathbf{y}), \mathbf{x} - \mathbf{y}\rangle \geq \alpha\|\nabla F(\mathbf{x}) - \nabla F(\mathbf{y})\|^2$ holds for any $\mathbf{x}, \mathbf{y} \in \mathcal{X}$ with parameter $\alpha > 0$. In our case, right side of the inequality will be replaced by $\alpha\|\nabla F(\mathbf{x}) - \nabla F(\mathbf{y})\|^2 - \varepsilon$, where $\varepsilon$ is a small.

Let us begin by defining the following two functions

$$\psi(\mathbf{W}) = \mathcal{L}_{S\backslash i}(\mathbf{W}) - \langle\nabla\mathcal{L}_{S\backslash i}(\mathbf{W}_t^{(i)}), \mathbf{W}\rangle\,, \qquad \psi^\star(\mathbf{W}) = \mathcal{L}_{S\backslash i}(\mathbf{W}) - \langle\nabla\mathcal{L}_{S\backslash i}(\mathbf{W}_t), \mathbf{W}\rangle\,.$$

Observe that

$$\psi(\mathbf{W}_t) - \psi(\mathbf{W}_t^{(i)}) + \psi^\star(\mathbf{W}_t^{(i)}) - \psi^\star(\mathbf{W}_t) \tag{18}$$
$$= \mathcal{L}_{S\backslash i}(\mathbf{W}_t) - \langle\nabla\mathcal{L}_{S\backslash i}(\mathbf{W}_t^{(i)}), \mathbf{W}_t\rangle - \mathcal{L}_{S\backslash i}(\mathbf{W}_t^{(i)}) + \langle\nabla\mathcal{L}_{S\backslash i}(\mathbf{W}_t^{(i)}), \mathbf{W}_t^{(i)}\rangle$$
$$+ \mathcal{L}_{S\backslash i}(\mathbf{W}_t^{(i)}) - \langle\nabla\mathcal{L}_{S\backslash i}(\mathbf{W}_t), \mathbf{W}_t^{(i)}\rangle - \mathcal{L}_{S\backslash i}(\mathbf{W}_t) + \langle\nabla\mathcal{L}_{S\backslash i}(\mathbf{W}_t), \mathbf{W}_t\rangle$$
$$= \langle\mathbf{W}_t - \mathbf{W}_t^{(i)}, \nabla\mathcal{L}_{S\backslash i}(\mathbf{W}_t) - \nabla\mathcal{L}_{S\backslash i}(\mathbf{W}_t^{(i)})\rangle\,,$$

from which follows that we are interesting in giving lower bounds on $\psi(\mathbf{W}_t) - \psi(\mathbf{W}_t^{(i)})$ and $\psi^\star(\mathbf{W}_t^{(i)}) - \psi^\star(\mathbf{W}_t)$.

From Lemma 1 we know the loss is $\rho$-smooth with $\rho = C_x^2 \left( B_{\phi'}^2 + B_{\phi''} B_\phi + \frac{C_y B_{\phi''}}{\sqrt{m}} \right)$, and thus, for any $i \in [n]$, we immediately have the upper bounds

$$\psi(\mathbf{W}_t - \nabla\psi(\mathbf{W}_t)) \leq \psi(\mathbf{W}_t) - \eta \left( 1 - \frac{\eta\rho}{2} \right) \|\nabla\psi(\mathbf{W}_t)\|_2^2 \tag{19}$$

$$\psi^\star(\mathbf{W}_t^{(i)} - \eta\nabla\psi^\star(\mathbf{W}_t^{(i)})) \leq \psi^\star(\mathbf{W}_t^{(i)}) - \eta \left( 1 - \frac{\eta\rho}{2} \right) \|\nabla\psi^\star(\mathbf{W}_t^{(i)})\|_2^2 \tag{20}$$

Now, in the smooth and convex case [Nesterov, 2003], convexity would be used here to lower bound the left side of each of the inequalities by $\psi(\mathbf{W}_t^{(i)})$ and $\psi^\star(\mathbf{W}_t)$ respectively. In our case, while the functions are not convex, we can get an "approximate" lower bound by leveraging that the minimum Eigenvalue evaluated at the points $\mathbf{W}_t, \mathbf{W}_t^{(i)}$ is not too small. More precisely, we have the following lower bounds by applying Lemma 7, which will be shown shortly:

$$\psi(\mathbf{W}_t - \eta\nabla\psi(\mathbf{W}_t)) \geq \psi(\mathbf{W}_t^{(i)}) - \frac{\epsilon}{2} \|\mathbf{W}_t - \mathbf{W}_t^{(i)} - \eta\nabla\psi(\mathbf{W}_t)\|_2^2 \,, \tag{21}$$

$$\psi^\star(\mathbf{W}_t^{(i)} - \eta\nabla\psi^\star(\mathbf{W}_t^{(i)})) \geq \psi^\star(\mathbf{W}_t) - \frac{\epsilon}{2} \|\mathbf{W}_t^{(i)} - \mathbf{W}_t - \eta\nabla\psi^\star(\mathbf{W}_t^{(i)})\|_2^2. \tag{22}$$

Combining this with Eq. (19), (20), and rearranging we get:

$$\psi(\mathbf{W}_t) - \psi(\mathbf{W}_t^{(i)}) \geq \eta \left( 1 - \frac{\eta\rho}{2} \right) \|\nabla\psi(\mathbf{W}_t)\|_2^2 - \frac{\epsilon}{2} \|\mathbf{W}_t - \mathbf{W}_t^{(i)} - \eta\nabla\psi(\mathbf{W}_t)\|_2^2 \,, \tag{23}$$

$$\psi^\star(\mathbf{W}_t^{(i)}) - \psi^\star(\mathbf{W}_t) \geq \eta \left( 1 - \frac{\eta\rho}{2} \right) \|\nabla\psi^\star(\mathbf{W}_t^{(i)})\|_2^2 - \frac{\epsilon}{2} \|\mathbf{W}_t^{(i)} - \mathbf{W}_t - \eta\nabla\psi^\star(\mathbf{W}_t^{(i)})\|_2^2. \tag{24}$$

Adding together the two bounds and plugging into Eq. (18) completes the proof.

**Proof of Eq. (21) and Eq. (22).** All that is left to do, is to prove Eq. (21) and (22). To do that, we will use Lemma 7 while recalling the definition of $\mathbf{W}(\alpha)$ and $\widetilde{\mathbf{W}}(\alpha)$ given in the Lemma. That said, let us then define the following two functions:

$$g(\alpha) \stackrel{\text{def}}{=} \psi(\mathbf{W}(\alpha)) + \frac{\widetilde{\epsilon}\alpha^2}{2} \|\mathbf{W}_t - \mathbf{W}_t^{(i)} - \eta\big(\nabla\mathcal{L}_{S\setminus i}(\mathbf{W}_t) - \nabla\mathcal{L}_{S\setminus i}(\mathbf{W}_t^{(i)})\big)\|_2^2 \,,$$

$$\widetilde{g}(\alpha) \stackrel{\text{def}}{=} \psi^\star(\widetilde{\mathbf{W}}(\alpha)) + \frac{\widetilde{\epsilon}\alpha^2}{2} \|\mathbf{W}_t - \mathbf{W}_t^{(i)} - \eta\big(\nabla\mathcal{L}_{S\setminus i}(\mathbf{W}_t) - \nabla\mathcal{L}_{S\setminus i}(\mathbf{W}_t^{(i)})\big)\|_2^2 \,.$$

Note that from Lemma 7 we have that $g''(\alpha), \widetilde{g}''(\alpha) \geq 0$ for $\alpha \in [0,1]$. Indeed, we have with $\Delta \stackrel{\text{def}}{=} \mathbf{W}_t - \mathbf{W}_t^{(i)} - \eta\big(\nabla\mathcal{L}_{S\setminus i}(\mathbf{W}_t) - \nabla\mathcal{L}_{S\setminus i}(\mathbf{W}_t^{(i)})\big)$:

$$g''(\alpha) = \big\langle \Delta, \nabla^2\mathcal{L}_{S\setminus i}(\mathbf{W}(\alpha))\Delta \big\rangle + \widetilde{\epsilon}\|\Delta\|_2^2 \geq 0$$

and similarly for $\widetilde{g}(\alpha)$. Therefore both $g(\cdot)$ and $\widetilde{g}(\cdot)$ are convex on $[0,1]$. The first inequality then arises from noting the follow three points. Since $g$ is convex we have $g(1) - g(0) \geq g'(0)$ with $g'(0) = \langle \nabla\psi(\mathbf{W}_t^{(i)}), \Delta \rangle = 0$ since $\nabla\psi(\mathbf{W}_t^{(i)}) = 0$. This yields

$$0 \leq g(1) - g(0)$$

$$= \psi(\mathbf{W}_t - \eta\nabla\psi(\mathbf{W}_t)) + \frac{\widetilde{\epsilon}}{2} \|\mathbf{W}_t - \mathbf{W}_t^{(i)} - \eta\big(\nabla\mathcal{L}_{S\setminus i}(\mathbf{W}_t) - \nabla\mathcal{L}_{S\setminus i}(\mathbf{W}_t^{(i)})\big)\|_2^2 - \psi(\mathbf{W}_t^{(i)})$$

which is almost Eq. (21): The missing step is showing that $\widetilde{\epsilon} \leq \epsilon$. This comes by the uniform boundedness of the loss, that is, having $\ell(\mathbf{W}_0, z) \leq C_0$ a.s. we can upper-bound

$$\widetilde{\epsilon} \leq 2 \cdot \frac{C_x^2 \sqrt{C_0} B_{\phi''}}{\sqrt{m}} \big( 4 B_{\phi'} C_x \sqrt{\eta s} + \sqrt{2} \big) = \epsilon$$

This proves Eq. (21), while Eq. (22) comes by following similar steps and considering $\widetilde{g}(1) - \widetilde{g}(0) \geq \widetilde{g}'(0)$.

### C.3.2    Proof of Lemma 6

Recalling the Hessian (11) we have for any parameter $\mathbf{W}$ and data point $z = (\mathbf{x}, y)$,

$$\|\nabla^2\ell(\mathbf{W}, z)\|_2 \leq \|\nabla f_{\mathbf{W}}(\mathbf{x})\|_2^2 + \|\nabla^2 f_{\mathbf{W}}(\mathbf{x})\|_2 |f_{\mathbf{W}}(\mathbf{x}) - y|$$

$$\leq C_x^2 \left( B_{\phi'}^2 + B_{\phi''} B_\phi + \frac{B_{\phi''} C_y}{\sqrt{m}} \right)$$

That is we have from (12) the bound $\|\nabla^2 f_{\mathbf{W}}(\mathbf{x})\|_2 \leq \frac{C_x^2}{\sqrt{m}} B_{\phi''}$, meanwhile we can trivially bound

$$|f_{\mathbf{W}}(\mathbf{x}) - y| \leq \frac{1}{\sqrt{m}} \sum_{j=1}^{m} |\phi(\langle (\mathbf{W})_j, \mathbf{x} \rangle)| + C_y$$

$$\leq \sqrt{m} B_\phi + C_y.$$

and

$$\|\nabla f_{\mathbf{W}}(\mathbf{x})\|_2^2 = \|\mathbf{x}\|_2^2 \cdot \frac{1}{m} \sum_{j=1}^{m} \phi'(\langle (\mathbf{W})_j, \mathbf{x} \rangle)$$

$$\leq C_x^2 B_{\phi'}^2.$$

The loss is therefore $\rho$-smooth with $\rho = C_x^2 \left( B_{\phi'}^2 + B_\phi B_{\phi''} + \frac{C_y B_{\phi''}}{\sqrt{m}} \right)$. Following standard arguments we then have for $j \in \mathbb{N}_0$

$$\mathcal{L}_S(\mathbf{W}_{j+1}) \leq \mathcal{L}_S(\mathbf{W}_j) - \eta \left( 1 - \frac{\eta\rho}{2} \right) \|\nabla \mathcal{L}_S(\mathbf{W}_j)\|_F^2$$

which when rearranged and summed over $j$ yields

$$\eta \left( 1 - \frac{\eta\rho}{2} \right) \sum_{j=0}^{t} \|\nabla \mathcal{L}_S(\mathbf{W}_j)\|_F^2 \leq \sum_{j=0}^{t} \mathcal{L}_S(\mathbf{W}_j) - \mathcal{L}_S(\mathbf{W}_{j+1}) = \mathcal{L}_S(\mathbf{W}_0) - \mathcal{L}_S(\mathbf{W}_{t+1})$$

We also note that

$$\mathbf{W}_{t+1} - \mathbf{W}_0 = -\eta \sum_{s=0}^{t} \nabla \mathcal{L}_S(\mathbf{W}_s)$$

and therefore by convexity of the squared norm we have $\|\mathbf{W}_{t+1} - \mathbf{W}_0\|_F^2 = \eta^2 \|\sum_{s=0}^{t} \nabla \mathcal{L}_S(\mathbf{W}_s)\|_F^2 \leq \eta^2 t \sum_{s=0}^{t} \|\nabla \mathcal{L}_S(\mathbf{W}_s)\|_F^2$. Plugging this in we get when $\eta\rho \leq 1/2$

$$\frac{3}{4} \cdot \frac{1}{\eta t} \|\mathbf{W}_{t+1} - \mathbf{W}_0\|_F^2 \leq \mathcal{L}_S(\mathbf{W}_0)$$

Rearranging then yields the inequality. An identical set of steps can be performed for the cases $\mathbf{W}_t^{(i)}$ for $i \in [n]$.

### C.3.3 Proof of Lemma 7

Looking at (11) we note the first matrix is positive semi-definite and therefore for any $\mathbf{W} \in \mathbb{R}^{dm}$:

$$\lambda_{\min}(\nabla^2 \mathcal{L}_{S\backslash i}(\mathbf{W})) \geq -\lambda_{\max} \left( \frac{1}{n} \sum_{j \in [n]: j \neq i} \nabla^2 f_{\mathbf{W}}(\mathbf{x}_i) \big( f_{\mathbf{W}}(\mathbf{x}_j) - y_j \big) \right)$$

$$\geq -\frac{C_x^2 B_{\phi''}}{\sqrt{m}} \cdot \frac{1}{n} \sum_{j \in [n]: j \neq i} |f_{\mathbf{W}}(x_j) - y_j|$$

where we have used the operator norm of the Hessian $\|\nabla^2 f_{\mathbf{W}}(\mathbf{x})\|_2$ bound (12). We now choose $\mathbf{W} = \mathbf{W}(\alpha)$ and thus need to bound $\frac{1}{n} \sum_{j \in [n]: j \neq i} |f_{\mathbf{W}(\alpha)}(x_j) - y_i|$ and $\frac{1}{n} \sum_{j \in [n]: j \neq i} |f_{\widetilde{\mathbf{W}}(\alpha)}(\mathbf{x}_i) - y_i|$. Note that we then have for any iterate $\mathbf{W}_t$ with $t \in \mathbb{N}_0$,

$$\frac{1}{n} \sum_{j \in [n]: j \neq i} |f_{\mathbf{W}(\alpha)}(\mathbf{x}_i) - y_i| \leq \frac{1}{n} \sum_{j \in [n]: j \neq i} |f_{\mathbf{W}(\alpha)}(\mathbf{x}_i) - f_{\mathbf{W}_t^{(i)}}(\mathbf{x}_i)| + \frac{1}{n} \sum_{j \in [n]: j \neq i} |f_{\mathbf{W}_t^{(i)}}(\mathbf{x}_i) - y_i|$$

$$\leq B_{\phi'} C_x \|\mathbf{W}(\alpha) - \mathbf{W}_t^{(i)}\|_F + \sqrt{2\mathcal{L}_{S\backslash i}(\mathbf{W}_t^{(i)})}$$

where the first term on the r.h.s. is bounded using Cauchy-Schwarz inequality as in Eq. (15)-(16), and the second term is bounded by Jensen's inequality. A similar calculation yields

$$\frac{1}{n} \sum_{j=1, j \neq i}^{n} |f_{\widetilde{\mathbf{W}}(\alpha)}(\mathbf{x}_i) - y_i| \leq B_{\phi'} C_x \|\widetilde{\mathbf{W}}(\alpha) - \mathbf{W}_t\|_F + \sqrt{2\mathcal{L}_{S\backslash i}(\mathbf{W}_t^{(i)})}.$$

Since the loss is $\rho$-smooth by Lemma 1 we then have

$$\|\mathbf{W}(\alpha) - \mathbf{W}_t^{(i)}\|_F \leq \alpha\big(\|\mathbf{W}_t - \mathbf{W}_t^{(i)}\|_F + \eta\|\nabla\mathcal{L}_{S\setminus i}(\mathbf{W}_t) - \nabla\mathcal{L}_{S\setminus i}(\mathbf{W}_t^{(i)})\|_F\big)$$
$$\leq (1 + \eta\rho)\|\mathbf{W}_t - \mathbf{W}_t^{(i)}\|_F$$
$$\leq \frac{3}{2}\big(\|\mathbf{W}_t - \mathbf{W}_0\|_F + \|\mathbf{W}_0 - \mathbf{W}_t^{(i)}\|_F\big)$$
$$\leq \frac{3}{2}\sqrt{2\eta s}\big(\sqrt{\mathcal{L}_S(\mathbf{W}_0)} + \sqrt{\mathcal{L}_{S^{(i)}}(\mathbf{W}_0)}\big)$$

where at the end we used Lemma 6. A similar calculation yields the same bound for $\|\widetilde{\mathbf{W}}(\alpha) - \mathbf{W}_t\|_F$. Bringing together we get

$$\lambda_{\min}(\nabla^2\mathcal{L}_{S\setminus i}(\mathbf{W}(\alpha))) \geq -\frac{C_x^2 B_{\phi''}}{\sqrt{m}}\left(4B_{\phi'}C_x\sqrt{\eta s}\big(\sqrt{\mathcal{L}_S(\mathbf{W}_0)} + \sqrt{\mathcal{L}_{S^{(i)}}(\mathbf{W}_0)}\big) + \sqrt{2\mathcal{L}_{S\setminus i}(\mathbf{W}_t^{(i)})}\right)$$

$$\lambda_{\min}(\nabla^2\mathcal{L}_{S\setminus i}(\widetilde{\mathbf{W}}(\alpha))) \geq -\frac{C_x^2 B_{\phi''}}{\sqrt{m}}\left(4B_{\phi'}C_x\sqrt{\eta s}\big(\sqrt{\mathcal{L}_S(\mathbf{W}_0)} + \sqrt{\mathcal{L}_{S^{(i)}}(\mathbf{W}_0)}\big) + \sqrt{2\mathcal{L}_{S\setminus i}(\mathbf{W}_t)}\right)$$

The final bound arises from noting that $\mathcal{L}_{S\setminus i}(\mathbf{W}_t) \leq \mathcal{L}_S(\mathbf{W}_t) \leq \mathcal{L}_S(\mathbf{W}_0)$ and $\mathcal{L}_{S\setminus i}(\mathbf{W}_t^{(i)}) \leq \mathcal{L}_{S^{(i)}}(\mathbf{W}_t^{(i)}) \leq \mathcal{L}_{S^{(i)}}(\mathbf{W}_0)$.

# D   Connection between $\Delta_S^{\mathrm{oracle}}$ and NTK

This section is dedicated to the proof of Theorem 2. We will first need the following standard facts about the NTK.

**Lemma 8** (NTK Lemma). *For any* $\mathbf{W}, \widetilde{\mathbf{W}} \in \mathbb{R}^{d\times m}$ *and any* $\mathbf{x} \in \mathbb{R}^d$,

$$f_{\mathbf{W}}(\mathbf{x}) = f_{\widetilde{\mathbf{W}}}(\mathbf{x}) + \sum_{k=1}^m u_k \phi'\left(\left\langle\mathbf{x}, \widetilde{\mathbf{W}}_k\right\rangle\right)\left\langle\mathbf{W}_k - \widetilde{\mathbf{W}}_k, \mathbf{x}\right\rangle + \epsilon(\mathbf{x})$$

*where*

$$\epsilon(\mathbf{x}) = \frac{1}{2}\sum_{k=1}^m u_k\left(\int_0^1 \phi''\left(\tau\left\langle\mathbf{x}, \mathbf{W}_k\right\rangle + (1-\tau)\left\langle\mathbf{x}, \widetilde{\mathbf{W}}_k\right\rangle\right)d\tau\right)\left\langle\mathbf{x}, \mathbf{W}_k - \widetilde{\mathbf{W}}_k\right\rangle^2.$$

*Note that*

$$|\epsilon(\mathbf{x})| \leq \frac{B_{\phi''}\|\mathbf{x}\|}{2\sqrt{m}} \cdot \|\mathbf{W} - \widetilde{\mathbf{W}}\|_F^2.$$

*Proof.* By Taylor theorem,

$$f_{\mathbf{W}}(\mathbf{x}) = f_{\widetilde{\mathbf{W}}}(\mathbf{x}) + \sum_k u_k \phi'\left(\left\langle\mathbf{x}, \widetilde{\mathbf{W}}_k\right\rangle\right)\left\langle\mathbf{x}, \mathbf{W}_k - \widetilde{\mathbf{W}}_k\right\rangle$$
$$+ \underbrace{\frac{1}{2}\sum_k u_k\left(\int_0^1 \phi''\left(\tau\left\langle\mathbf{x}, \mathbf{W}_k\right\rangle + (1-\tau)\left\langle\mathbf{x}, \widetilde{\mathbf{W}}_k\right\rangle\right)d\tau\right)\left\langle\mathbf{x}, \mathbf{W}_k - \widetilde{\mathbf{W}}_k\right\rangle^2}_{\epsilon(\mathbf{x})}.$$

Cauchy-Schwarz inequality gives us

$$|\epsilon(\mathbf{x})| \leq \frac{B_{\phi''}\|\mathbf{x}\|}{2\sqrt{m}} \cdot \|\mathbf{W} - \widetilde{\mathbf{W}}\|_F^2.$$

$\square$

We will use the following proposition [Du et al., 2018, Arora et al., 2019]:

**Proposition 1** (Concentration of NTK gram matrix). *With probability at least $1 - \delta$ over $\mathbf{W}_0$,*

$$\|\hat{\mathbf{K}} - \mathbf{K}\|_2 \leq B_{\phi'} \sqrt{\frac{\ln\left(\frac{2n}{\delta}\right)}{2m}} \ .$$

*Proof.* Since each entry is independent, by Hoeffding's inequality we have for any $t \geq 0$,

$$\mathbf{P}\left(n|(\hat{\mathbf{K}})_{i,j} - (\mathbf{K})_{i,j}| \geq t\right) \leq 2e^{-2nt^2/B_{\phi'}^2} \ ,$$

and applying the union bound

$$\|\hat{\mathbf{K}} - \mathbf{K}\|_F^2 \leq \frac{B_{\phi'}^2 \ln\left(\frac{2n}{\delta}\right)}{2m} \ .$$

$\square$

Now we are ready to prove the main Theorem of this section (in the main text we only report the second result).

**Theorem 2 (restated).** *Denote*

$$\mathbf{\Phi}_0 \overset{\text{def}}{=} \begin{bmatrix} u_1 \mathbf{X} \operatorname{diag}\left(\phi'(\mathbf{X}^\top \mathbf{W}_{0,1})\right) \\ \vdots \\ u_m \mathbf{X} \operatorname{diag}\left(\phi'(\mathbf{X}^\top \mathbf{W}_{0,m})\right) \end{bmatrix}$$

*and $\hat{\mathbf{K}} \overset{\text{def}}{=} \frac{1}{n}\mathbf{\Phi}_0^\top \mathbf{\Phi}_0$. Assume that $m \gtrsim (\eta T)^5$. Then,*

$$\Delta_S^{\text{oracle}} = \mathcal{O}\left(\frac{1}{\eta T}\left\langle \mathbf{y}, (n\hat{\mathbf{K}})^{-1}\mathbf{y}\right\rangle\right) \quad \text{as} \quad \eta T \to \infty \ .$$

*Consider Assumption 1 and that $\eta T = n$. Moreover, assume that entries of $\mathbf{W}_0$ are i.i.d., $\mathbf{K} = \mathbf{E}[\hat{\mathbf{K}} \mid S, \mathbf{u}]$ with $\lambda_{\min}(\mathbf{K}) \gtrsim 1/n$, and assume that $\mathbf{u} \sim \operatorname{unif}\left(\{\pm 1/\sqrt{m}\}\right)^m$ independently from all sources of randomness. Then, with probability least $1 - \delta$ over $(\mathbf{W}_0, \mathbf{u})$,*

$$\Delta_S^{\text{oracle}} = \tilde{\mathcal{O}}_P\left(\frac{1}{n}\left\langle \mathbf{y}, (n\mathbf{K})^{-1}\mathbf{y}\right\rangle\right) \quad \text{as} \quad n \to \infty \ .$$

*Proof.* The proof of the first inequality will follow by relaxation of the oracle R-ERM $\Delta_S^{\text{oracle}}$ to the Moore-Penrose pseudo-inverse solution to a linearised problem given by Lemma 8. The proof of the second inequality will build on the same idea, in addition making use of the concentration of entries of $\hat{\mathbf{K}}$ around $\mathbf{K}$.

Define

$$f_{\mathbf{W}}^{\text{lin}}(\mathbf{x}) \overset{\text{def}}{=} \sum_{k=1}^m u_k \phi'\left(\langle \mathbf{x}, \mathbf{W}_{0,k}\rangle\right)\langle \mathbf{W}_k - \mathbf{W}_{0,k}, \mathbf{x}\rangle \ ,$$

$$\mathcal{L}_S^{\text{lin}}(\mathbf{W}) \overset{\text{def}}{=} \frac{1}{2}\sum_{i=1}^n \left(y_i - f_{\mathbf{W}}^{\text{lin}}(\mathbf{x})\right)^2 \ .$$

Then for the square loss we have

$$\begin{aligned} (f_{\mathbf{W}}(\mathbf{x}_i) - y_i)^2 &= \left(f_{\mathbf{W}_0}(\mathbf{x}_i) + f_{\mathbf{W}}^{\text{lin}}(\mathbf{x}_i) + \epsilon(\mathbf{x}_i) - y_i\right)^2 \\ &\leq 2\left(f_{\mathbf{W}}^{\text{lin}}(\mathbf{x}_i) - (y_i - f_{\mathbf{W}_0}(\mathbf{x}_i))\right)^2 + 2\epsilon(\mathbf{x}_i)^2 \end{aligned}$$

and so,

$$\mathcal{L}(\mathbf{W}) \leq \mathcal{L}^{\text{lin}}(\mathbf{W}) + \frac{B_{\phi''}^2}{m}\cdot\|\mathbf{W} - \mathbf{W}_0\|_F^4$$

where we observe that

$$\mathcal{L}^{\mathrm{lin}}(\mathbf{W}) = \frac{1}{n}\|\mathbf{\Phi}_0^\top(\mathbf{W} - \mathbf{W}_0) - (\mathbf{y} - \hat{\mathbf{y}}_0)\|^2$$

and with $\mathbf{\Phi}_0$, the matrix of NTK features, defined in the statement.

Solving the above undetermined least-squares problem using the Moore-Penrose pseudo-inverse we get

$$\mathbf{W}^{\mathrm{pinv}} - \mathbf{W}_0 = \left(\mathbf{\Phi}_0\mathbf{\Phi}_0^\top\right)^\dagger \mathbf{\Phi}_0(\mathbf{y} - \hat{\mathbf{y}}_0)\,,$$

and so

$$\begin{aligned}
\|\mathbf{W}^{\mathrm{pinv}} - \mathbf{W}_0\|_F^2 &= (\mathbf{y} - \hat{\mathbf{y}}_0)^\top \mathbf{\Phi}_0^\top \left(\mathbf{\Phi}_0\mathbf{\Phi}_0^\top\right)^{\dagger 2} \mathbf{\Phi}_0(\mathbf{y} - \hat{\mathbf{y}}_0)\\
&= (\mathbf{y} - \hat{\mathbf{y}}_0)^\top \left(\mathbf{\Phi}_0^\top \mathbf{\Phi}_0\right)^{-1} (\mathbf{y} - \hat{\mathbf{y}}_0)\\
&= (\mathbf{y} - \hat{\mathbf{y}}_0)^\top (n\hat{\mathbf{K}})^{-1}(\mathbf{y} - \hat{\mathbf{y}}_0)
\end{aligned}$$

where the final step can be observed by Singular Value Decomposition (SVD) of $\mathbf{\Phi}_0$. Since $\mathcal{L}_S(\mathbf{W}^{\mathrm{pinv}}) = 0$,

$$\Delta_S^{\mathrm{oracle}} = \mathcal{O}\left(\frac{1}{\eta T}\left\langle(\mathbf{y} - \hat{\mathbf{y}}_0), (n\hat{\mathbf{K}})^{-1}(\mathbf{y} - \hat{\mathbf{y}}_0)\right\rangle\right) \quad \text{as} \quad \eta T \to \infty\,.$$

This proves the first result.

Now we prove the second result involving $\mathbf{K}$. We will first handle the empirical risk by concentration between $\hat{\mathbf{K}}$ and $\mathbf{K}$. For $\boldsymbol{\alpha} \in \mathbb{R}^n$ define $\mathbf{W}_{\boldsymbol{\alpha}} = \mathbf{\Phi}_0\boldsymbol{\alpha} + \mathbf{W}_0$. Then,

$$\begin{aligned}
\mathcal{L}^{\mathrm{lin}}(\mathbf{W}_{\boldsymbol{\alpha}}) &= \frac{1}{n}\|\mathbf{\Phi}_0^\top\mathbf{\Phi}_0\boldsymbol{\alpha} - (\mathbf{y} - \hat{\mathbf{y}}_0)\|^2\\
&= \frac{1}{n}\|n(\hat{\mathbf{K}} - \mathbf{K})\boldsymbol{\alpha} + n\mathbf{K}\boldsymbol{\alpha} - (\mathbf{y} - \hat{\mathbf{y}}_0)\|^2\\
&\leq \frac{2}{n}\|n(\hat{\mathbf{K}} - \mathbf{K})\boldsymbol{\alpha}\|^2 + \frac{2}{n}\|n\mathbf{K}\boldsymbol{\alpha} - (\mathbf{y} - \hat{\mathbf{y}}_0)\|^2\\
&\leq 2n\|\hat{\mathbf{K}} - \mathbf{K}\|_2^2\|\boldsymbol{\alpha}\|_2^2 + \frac{2}{n}\|n\mathbf{K}\boldsymbol{\alpha} - (\mathbf{y} - \hat{\mathbf{y}}_0)\|^2
\end{aligned}$$

Plug into the above $\hat{\boldsymbol{\alpha}} = (n\mathbf{K})^{-1}(\mathbf{y} - \hat{\mathbf{y}}_0)$ (note that $\mathbf{K}$ is full-rank by assumption)

$$\begin{aligned}
\mathcal{L}^{\mathrm{lin}}(\mathbf{W}_{\hat{\boldsymbol{\alpha}}}) &\leq 2n\|\hat{\mathbf{K}} - \mathbf{K}\|_2^2\|\hat{\boldsymbol{\alpha}}\|_2^2\\
&\leq n \cdot \frac{B_{\phi'}^2 \ln\left(\frac{2n}{\delta}\right)}{m} \cdot \left((\mathbf{y} - \hat{\mathbf{y}}_0)^\top(n\mathbf{K})^{-2}(\mathbf{y} - \hat{\mathbf{y}}_0)\right)\\
&\leq \|\mathbf{y} - \hat{\mathbf{y}}_0\|^2 \cdot \frac{B_{\phi'}^2 \ln\left(\frac{2n}{\delta}\right)}{m} \cdot \frac{1}{n\lambda_{\min}(\mathbf{K})^2}\\
&= 2\mathcal{L}_S(\mathbf{W}_0) \cdot \frac{B_{\phi'}^2 \ln\left(\frac{2n}{\delta}\right)}{m} \cdot \frac{1}{\lambda_{\min}(\mathbf{K})^2}
\end{aligned}$$

where the last inequality hold w.p. at least $1 - \delta$ by Proposition 1.

Now we pay attention to the quadratic term within $\Delta_S^{\mathrm{oracle}}$:

$$\begin{aligned}
\|\mathbf{W}_{\hat{\boldsymbol{\alpha}}} - \mathbf{W}_0\|_2^2 &= \|\mathbf{\Phi}_0\hat{\boldsymbol{\alpha}}\|_2^2\\
&= \|\mathbf{\Phi}_0(n\mathbf{K})^{-1}(\mathbf{y} - \hat{\mathbf{y}}_0)\|_2^2\\
&= (\mathbf{y} - \hat{\mathbf{y}}_0)^\top(n\mathbf{K})^{-1}(n\hat{\mathbf{K}})(n\mathbf{K})^{-1}(\mathbf{y} - \hat{\mathbf{y}}_0)\\
&= \underbrace{(\mathbf{y} - \hat{\mathbf{y}}_0)^\top(n\mathbf{K})^{-1}(n\hat{\mathbf{K}} - n\mathbf{K})(n\mathbf{K})^{-1}(\mathbf{y} - \hat{\mathbf{y}}_0)}_{(i)} + \underbrace{(\mathbf{y} - \hat{\mathbf{y}}_0)^\top(n\mathbf{K})^{-1}(\mathbf{y} - \hat{\mathbf{y}}_0)}_{(ii)}\,.
\end{aligned}$$

We will show that $(i)$ is "small":

$$(\mathbf{y} - \hat{\mathbf{y}}_0)^\top (n\mathbf{K})^{-1}(n\hat{\mathbf{K}} - n\mathbf{K})(n\mathbf{K})^{-1}(\mathbf{y} - \hat{\mathbf{y}}_0)$$

$$\leq \|\mathbf{y} - \hat{\mathbf{y}}_0\|^2 \|(n\mathbf{K})^{-2}\| n \|\hat{\mathbf{K}} - \mathbf{K}\|_2$$

$$\leq \|\mathbf{y} - \hat{\mathbf{y}}_0\|^2 \|(n\mathbf{K})^{-2}\| \cdot n B_{\phi'} \sqrt{\frac{\ln\left(\frac{2n}{\delta}\right)}{2m}}$$

$$\leq 2\mathcal{L}_S(\mathbf{W}_0) \cdot \frac{1}{\lambda_{\min}(\mathbf{K})^2} \cdot B_{\phi'} \sqrt{\frac{\ln\left(\frac{2n}{\delta}\right)}{2m}}$$

where we used Proposition 1 once again. Putting all together w.p. at least $1 - \delta$ over $\mathbf{W}_0$ we have

$$\Delta_S^{\text{oracle}} = \mathcal{O}_P \left( \frac{1}{\eta T} \left\langle (\mathbf{y} - \hat{\mathbf{y}}_0), (n\mathbf{K})^{-1}(\mathbf{y} - \hat{\mathbf{y}}_0) \right\rangle \right.$$

$$\left. + \frac{2\mathcal{L}_S(\mathbf{W}_0)}{\lambda_{\min}(\mathbf{K})^2} \cdot \frac{B_{\phi'}^2 \ln\left(\frac{2n}{\delta}\right)}{m} + \frac{1}{\eta T} \cdot \frac{2\mathcal{L}_S(\mathbf{W}_0)}{\lambda_{\min}(\mathbf{K})^2} \cdot B_{\phi'} \sqrt{\frac{\ln\left(\frac{2n}{\delta}\right)}{2m}} \right) \quad \text{as} \quad \eta T \to \infty .$$

Moreover, assuming that $\lambda_{\min}(\mathbf{K}) \gtrsim 1/n$ and $\eta T = n$, the above turns into

$$\Delta_S^{\text{oracle}} = \tilde{\mathcal{O}}_P \left( \frac{1}{n} \left\langle (\mathbf{y} - \hat{\mathbf{y}}_0), (n\mathbf{K})^{-1}(\mathbf{y} - \hat{\mathbf{y}}_0) \right\rangle \right) \quad \text{as} \quad n \to \infty .$$

The final bit is to note that

$$\left\langle \hat{\mathbf{y}}_0, (n\mathbf{K})^{-1}\hat{\mathbf{y}}_0 \right\rangle \leq \frac{\|\hat{\mathbf{y}}_0\|_2^2}{n\lambda_{\min}(\mathbf{K})} \lesssim \|\hat{\mathbf{y}}_0\|_2^2$$

can be bounded w.h.p. by randomising $\mathbf{u} \sim \text{unif}\left(\{\pm 1/\sqrt{m}\}\right)^m$: For any $i \in [n]$ and $\delta \in (0,1)$ by Hoeffding's inequality we have:

$$\mathbf{P}\left( f_{\mathbf{W}_0}(\mathbf{x}_i) \geq B_\phi \sqrt{\frac{\ln\left(\frac{1}{\delta}\right)}{2}} \right) \geq \mathbf{P}\left( f_{\mathbf{W}_0}(\mathbf{x}_i) \geq \sqrt{\frac{\ln\left(\frac{1}{\delta}\right)}{2} \frac{1}{m} \sum_{k=1}^m \phi\left(\langle (\mathbf{W}_0)_k, \mathbf{x}\rangle\right)^2} \right)$$

$$\geq 1 - \delta .$$

Taking a union bound over $i \in [n]$ completes the proof of the second result. $\qquad \square$

# E    Additional Proofs

**Corollary 2 (restated).** *Assume the same as in Theorem 1 and Lemma 2. Then,*

$$\mathbf{E}\left[\mathcal{L}(\mathbf{W}_T) \mid \mathbf{W}_0, \mathbf{u}\right] \leq \left(1 + C \cdot \frac{\eta T}{n}\left(1 + \frac{\eta T}{n}\right)\right)\mathbf{E}\left[\Delta_S^{\text{oracle}} \mid \mathbf{W}_0, \mathbf{u}\right] \ .$$

*Proof.* Considering Theorem 1 with $t = T - 1$, and noting that $\mathcal{L}_S(\mathbf{W}_T) \leq \frac{1}{T}\sum_{j=0}^{T}\mathcal{L}_S(\mathbf{W}_j)$ then yields

$$\mathbf{E}[\mathcal{L}(\mathbf{W}_T) \mid \mathbf{W}_0, \mathbf{u}] \leq \left(1 + b\left(\frac{\eta T}{n} + \frac{\eta^2 T^2}{n^2}\right)\right)\frac{1}{T}\sum_{j=0}^{T}\mathbf{E}[\mathcal{L}_S(\mathbf{W}_j) \mid \mathbf{W}_0, \mathbf{u}]$$

$$\leq \left(1 + b\left(\frac{\eta T}{n} + \frac{\eta^2 T^2}{n^2}\right)\right)$$

$$\cdot \mathbf{E}\left[\min_{\mathbf{W}\in\mathbb{R}^{d\times m}}\left\{\mathcal{L}_S(\mathbf{W}) + \frac{\|\mathbf{W} - \mathbf{W}_0\|_F^2}{\eta t} + \frac{\widetilde{b}}{\sqrt{m}}\cdot\frac{1}{T}\sum_{j=0}^{T}\left(1 \vee \|\mathbf{W} - \mathbf{W}_j\|_F\right)^3\right\} \,\middle|\, \mathbf{W}_0, \mathbf{u}\right]$$

where at the end we applied Lemma 2 to bound $\frac{1}{T}\sum_{j=0}^{T}\mathbf{E}[\mathcal{L}_S(\mathbf{W}_j)|\mathbf{W}_0]$. The constants $b, \widetilde{b}$ are then defined in Theorem 1 and Lemma 2. Note from smoothness of the loss we have

$$\|\mathbf{W} - \mathbf{W}_j\|_F^3 \leq 2^{3/2}\left(\|\mathbf{W} - \mathbf{W}_0\|_F^3 + \|\mathbf{W}_0 - \mathbf{W}_j\|_F^3\right) \leq 2^{3/2}\left(\|\mathbf{W} - \mathbf{W}_0\|_F^3 + (\eta j C_0)^{3/2}\right),$$

in particular from the properties of graident descent $\|\mathbf{W}_0 - \mathbf{W}_j\|_F^2 \leq \eta j \mathcal{L}_S(\mathbf{W}_0)$ for $j \in [T]$. Plugging in then yields the final bound. $\qquad\square$