# OpenReview forum: "Stability & Generalisation of Gradient Descent for Shallow Neural Networks without the Neural Tangent Kernel"
_NeurIPS.cc/2021/Conference — NeurIPS 2021 Poster_

### Official Review · Reviewer_9Lej · 2021-07-02

**Rating:** 6
**Confidence:** 3

**Summary:**

The authors propose to apply stability-based bound on the overparameterized neural networks and reach non-vacuous bound with early stopping. The authors show that the excess risk can be bounded by an interpolating network with the shortest GD path from initialization. The proof can be extended to NTK regimes and brings some interesting conclusions, e.g., GD with early stopping is consistent. Overall, I think this paper is interesting, and I tend to accept this paper.

**Ethical Concerns:**

No ethical issues.

**Limitations And Societal Impact:**

No potential negative societal impact.

**Main Review:**

【contributions】

(a.) The authors propose to apply stability-based bound on the overparameterized neural networks and show that the excess risk can be bounded by an interpolating network with the shortest GD path from initialization (Theorem 1).

(b.) There are some interesting conclusions based on the analysis. For example, (1) GD with early stopping is consistent; (2) one can recover the NTK-form generalization gap via early stopping (Theorem 2). (The early-stopping requirements, in my opinion, are the intrinsic weakness of stability-based bound.)

(c.) The proof is clear and easy to follow. The authors provide some intuitions behind the proof and the theorems. I enjoy reading such a paper.


【questions】

(a)   Line 20, maybe I missed some details. Why do we need |y| = C a constant?

(b)   Does the results of Arora et al. require \eta T = n? If not, I do not think the claim “our analysis is general enough to recover the results of Arora et al.” is fair.

【some typos】

(a.)  The formula between Line 244-245 lacks “+”

(b.) The motivation of the paper can be significantly improved. I am a little puzzled about whether the authors want to do optimization or generalization when reading the introduction. For example, Line 40-41, “thus, it is not clear whether GD can minimize it up to the desired precision,” makes me think the authors want to do optimization.

(c.)  Maybe I lose some details. Line 101, the condition is m>T^5, while in Theorem 1, the condition is T^3. Could the authors clarify this mismatching?

**Time Spent Reviewing:**

6

---

> ### Author Response · Authors · 2021-08-09
> **Comment to Reviewer Four**
>
> We provide answers to specific questions in the following bullet point format.
> * Typo  $|y| = C$ instead of  $|y| \leq C$.
>     -  It is sufficient to have $|y| \leq C$, not equal to constant. This is updated within the manuscript.
>
> * Does the results of Arora et al. require $\eta T = n$?
>     - Yes, as discussed within the introduction of our work, the results of Arora et al. require $T \geq 1/(\eta \lambda_{\min}(\mathbf{K}))$, and thus, applying that  $\lambda_{\min}(\mathbf{K}) \gtrsim 1/n$ from [Bartlett et. al. 2019] gives $\eta T \approx n$.
>
> * Manuscript motivation: Optimistion and Generalisation
>     - One contribution of our work is to show, for two layer neural networks, that optimisation controls generalisation (see Theorem 1). Given this, by obtaining guarantees on the optimisation error, we then obtain guarantees on the generalisation error. This connection is now more clearly described within the introduction.
>
> * Arriving at the requirement $m \geq T^5$
>     - This arises to ensure the additive factor $(\eta t)^{3/2}/\sqrt{m} $ in  the Optimisation error of  Lemma 1 is $O(1/(\eta t))$. In particular, this is ensured when $m \geq (\eta t)^{5}$.

---

> ### Author Response · Authors · 2021-09-06
> **Follow up to Reviewer Four**
>
> Dear Reviewer,
>
> Just a quick follow up regarding our response to your comments.
> Specifically, If you have any other outstanding concerns, do let us know as we can provide additional clarification.
>
> Otherwise, if you have no other concerns, we would kindly ask that you update your review in light of this.
>
> Best,
> Authors.

---

### Official Review · Reviewer_8wG3 · 2021-07-12

**Rating:** 6
**Confidence:** 3

**Summary:**

This paper provides a novel generalization guarantee for over-parameterized one-hidden-layer neural network by using oracle-based stability analysis instead of NTK.

**Limitations And Societal Impact:**

The limitations of the theorems are addressed in the paper. Since this is a theory paper, the societal impact can be negligible.

**Main Review:**

The paper provides a new type of non-vacuous generalization bound for over-parameterized shallow neural network. It's original and novel and may provide a new direction for generalization guarantees for over-parameterized models.

Most parts of the paper is easy to follow. The organization of the paper could be improved, such as moving main results section earlier so that we can remove some equation/statement duplication in both Sec. 2 and Sec. 3, such as the duplication of Eq. (4) and Corollary 1.

Line 154 mentioned that the oracle-based bound in this paper is tighter than NTK-based bounds because some relaxation of oracle-based bound is applied to achieve an equivalence to NTK-based bound. I am wondering if the authors can provide some examples showing the tighter-ness. For example, does there exist a target function such that oracle-based bound is non-vacuous while NTK-based bound is vacuous?

A few proof steps are not clear to me. For example, in footnote 3 in Page 3, why $L_S (W_T) \lesssim exp(\mu\sum_{j=1}^T 1/j)$ ? Also, I didn't find the proof of Corollary 1 and it's not straight forward to me that Corollary 1 can be derived directly from Theorem 1 and Lemma 1.

Overall, I believe the paper and its techniques will be interesting to the deep learning theory community.

**Time Spent Reviewing:**

4

---

> ### Author Response · Authors · 2021-08-09
> **Comment to Reviewer Three**
>
> We provide answers to specific questions in the following bullet point format.
> * Derivation of $L(W_{T}) \lesssim \exp(\mu \sum_{j=1}^{T} 1/j)$
>     - For a loss that is both smooth and $\mu-$Polyak-Łojasiewicz, we have
>      after $T$ steps of gradient descent with stepsizes $\eta_{1},\eta_{2},\dots,\eta_{T}$ the bound  $ L(W_{T}) \leq (1- \eta_{T} \mu) L (W_{T-1}) \leq L(W_{0}) \prod_{s=T}^{1} ( 1- \eta_{s} \mu) $. Applying $1-x \leq e^{-x}$ as well as $\eta_s = 1/s$ we then get $\prod_{s=T}^{1} ( 1- \eta_s\mu) \leq \exp(\mu \sum_{s=1}^{T}\eta_s) = \exp(\mu \sum_{s=1}^{T} 1/s)$. These details are now included within the manuscript.
>
> * Clarity of proof for Corollary 1 from Theorem 1 and Lemma 1.
>     -  We have now included greater details within the proof of Corollary 1. In particular, noting that the result is arrived at by multiplying and dividing by $t$ the right side of Theorem 1, and plugging in the oracle bound from Lemma 1.

---

> ### Author Response · Authors · 2021-09-06
> **Follow up to Reviewer Three**
>
> Dear Reviewer,
>
> Just a quick follow up regarding our response to your comments.
> Specifically, If you have any other outstanding concerns, do let us know as we can provide additional clarification.
>
> Otherwise, if you have no other concerns, we would kindly ask that you update your review in light of this.
>
> Best,
> Authors.

---

> ### Comment · Reviewer_8wG3 · 2021-09-06
> **Thank you for the response.**
>
> Thank you for the response. The authors' response clarify my question. I'll keep my score. Thank you.

---

### Official Review · Reviewer_EoLw · 2021-07-16

**Rating:** 4
**Confidence:** 4

**Summary:**

This paper derives bounds for generalization gap of two-layer neural networks using algorithmic stability. The bounds covers noisy and noiseless cases, and depend on the distance of the interpolation solution which is closest to the initialization. With properly chosen number of iterations and network width (with respect to the number of data n), the bounds decay as n increases.

The bounds requires the width of the networks to be bigger than a polynomial of the number of iterations (specifically, number of iterations times the step size). Hence, it works out of the NTK regime. Through, the bounds can also yield reasonable results in the NTK regime.

**Limitations And Societal Impact:**

The authors have adequately addressed the limitations of the work. Since this is a pure theoretical work, there's no potential negative societal impact.

**Main Review:**

This paper derives generalization bounds which covers broader cases than the NTK. Though, the assumptions for the network seems to be stronger than NKT. For example, the output parameters u are fixed, and the activation function is smooth and bounded. Under these assumptions, bounding the Hessians of the loss function becomes easy, and thus bounding the stability of GD also becomes easy. However, with the assumptions no special structure or property of neural network models is employed in the analysis, and the same technique can hardly be extended to deep neural networks without producing exponential terms of the depth.

Other major comments:
1. Although the bounds work out of the NTK regime, but since the width m should be greater than (\eta T)^5, it can easily fall into the NTK regime. As is mentioned in the paper, taking T=n^\alpha, then m>n^{5\alpha}. When \alpha>=0.5, the case is already in the NTK regime. So it seems the new bounds do not extend much from the NTK.

2. The deduction from equation 4 to the equation between line 106 and 107 looks confusing to me. Replacing E\Delta with equation 3 gives ||W-W_0||/(\eta*T), this cancels with the (\eta*T) on the denominator on the right hand side of equation 4, and leaves a 1/n. Why the result has 1/n^\alpha instead of 1/n?

3. It seems the proof part, Section 3, is written in a rush. There are many hard-to-read sentences and typos. For example, in line 198 "The activation \phi(u) continuous and twice differentiable ", and in line 205 "the main theorem a bound on for the gradient descent iterates", and line 211-213 "the generalisation gap can be controlled through both: the step size and number of iterations through the factor alongside the trajectory taken by gradient through the Empirical Risk".

**Time Spent Reviewing:**

4h

---

> ### Author Response · Authors · 2021-08-09
> **Comment to Reviewer Two**
>
> We provide answers to specific questions in the following bullet point format.
> + Comparison to Neural Tangent Kernel
>     - See response **Comparison to Neural Tangent Kernel (NTK)** in Comment to all Reviewers.
>
> + Comment "*... the assumptions for the network seems to be stronger than NKT... the output parameters u are fixed, and the activation function is smooth and bounded*".
>     - Fixing the output parameters is standard in the analysis of two layer neural networks (Arora et al., 2019), and provide the most natural first step towards understanding the non-convex nature of neural networks. NTK proofs work well with unbounded activations because they require randomisation of initial parameters (regardless of boundedness) --- our proof does not require this, and as such our assumptions are weaker. However, our technique is perfectly extendable to unbounded activations: Please also see a Comment to all Reviewers.
>
> + Comment "*Under these assumptions, bounding the Hessians of the loss function becomes easy, and thus bounding the stability of GD also becomes easy.*"
>     - To the best of our knowledge, this technique is new to the neural network algorithmic stability literature, and we believe that its intuitive simplicity is an advantage. On a technical level, bounding the Hessian is non-trivial as it requires its evaluation along a line segment $\mathbf{W}(\alpha)$ (see line 249 in the proof sketch Section 3 of the manuscript). This line (informally) joins the gradient descent iterates with and without a resampled data point.
>
> + Comment "*... no special structure or property of neural network models is employed in the analysis.*"
>     -  Our proof works $\textit{directly}$ with gradients and Hessians of the loss given a shallow neural network, and therefore exploits its structure in full.  Specifically, we use the problem structure to bound the Hessian along the line segment $\mathbf{W}(\alpha)$  (see Lemma 6 in Supplementary material), with the lower bound $\epsilon$ depending on the loss at initialisation and the activation's smoothness.
>
> + Comment "*... same technique can hardly be extended to deep neural networks without producing exponential terms of the depth* ".
>     -  The minimum Eigenvalue of a Hessian associated to deeper networks has been observed to be small both empirically (Figure 2 within [2]) as well as  theoretically [3] and (Richards and Rabbat, 2021). As such, it is reasonable that the techniques developed in our work can be extended to the deeper networks. We would be interested to hear reviewer's thoughts on why this is hardly the case.
>     - $[2]$ - Stagewise Training Accelerates Convergence of Testing Error Over SGD  Zhuoning Yuan, Yan Yan, Rong Jin, Tianbao Yang NIPS 2019
>     - $[3]$ - Loss landscapes and optimization in over-parameterized non-linear systems and neural networks Chaoyue Liu, Libin Zhu, and Mikhail Belkin. ArXiv 2020
>
> + Why result follow equation (4) has $1/n^{\alpha}$ not $1/n$
>     - We thank the reviewer for highlighting this typo. Indeed, in the noiseless case the Generalisation Error is $\frac{1}{n} ||\widehat{\mathbf{W}} - \mathbf{W}_0||_2^2  =  \frac{1}{n} \langle \mathbf{y} , (n \mathbf{K})^{-1} \mathbf{y}\rangle $ where $\widehat{\mathbf{W}}$ is the  *minimal-norm interpolating network*, and thus, recovers [Arora et. al., 2019]. This typo has now been fixed within the manuscript.
>
> + Readability of Section 3 (proof sketch).
>     - We thank you for the feedback regarding the written structure of section 3. Steps have been made to improve its readability and provide more clarity on the key steps. This includes dedicating more space to calculations and referencing more steps from the supplementary material.

---

> > ### Comment · Reviewer_EoLw · 2021-08-30
> > **Response for the comments**
> >
> > Thank you for the response. Most of my questions get addressed. Though, from my perspective the authors still did not clarify the preponderance of the proposed analysis over NTK. On this point, I agree with reviewer Rrma that "an explicit nontrivial theoretical example where the current paper gives an explicit nontrivial bound while NTK cannot" will be persuasive.
> >
> > For the extension to deep networks. It is indeed easy to write a bound. My point is, usually the bound will depend on an exponential term of the depth, which is not good (though empirically things may not be so bad). And it seems there's no good way to get rid of this exponential dependency of depth.

---

> > > ### Author Response · Authors · 2021-08-30
> > > **Cases where the current paper gives an explicit nontrivial bound while NTK cannot**
> > >
> > > Thank you for your response.
> > >
> > > In response to _"...from my perspective the authors still did not clarify the preponderance of the proposed analysis over NTK."_, please see below (we also provided this response to reviewer Rrma).
> > > We provide several _factual_ examples where the paper gives a non-trivial bound where NTK does not.
> > > If this does not qualify in your opinion, we would like to understand why and it would be helpful to get a _concrete_ and _constructive_ feedback.
> > >
> > >
> > > About the exponential dependence on depth: Is it really avoidable?
> > > It is known that bounds on the generalization error can be pseudo-linear in depth, e.g. _"Golowich,  Rakhlin, and Shamir. Size-independent sample complexity of neural networks." COLT 2018"_, **however** whether this is possible for the optimization error is unclear.
> > > NTK framework surely suffers from this.
> > > Therefore, not only we find that the depth criticism is tangential to our paper, but also might be unjustified in general.
> > >
> > >
> > > **Clarifications about cases where the current paper gives an explicit nontrivial bound while NTK cannot**:
> > >
> > > + We show a trade-off between overparameterisation and convergence rate, $n^{5 \alpha}$ vs $n^{-\alpha}$ respectively. We are not aware of any NTK results capable of this flexibility. One can choose $\alpha$ small enough to break a given polynomial overparameterisation assumption (e.g. Arora et al.) and still have a convergence.
> > >
> > > + We show consistency in the presence of label noise. We are not aware of any published NTK results showing this in a parametric setting. Note our bounds *do not* introduce any unknown quantities nor require tuning w.r.t. to them to achieve this: To show this in NTK, one could exploit early stopping results in kernel literature such as [1], however such bounds would depend on unknown quantities such as spectral profile of a kernel.
> > >
> > > + Our proof does not require randomised initialisation, which is critical for NTK. Our bounds are therefore strictly more general, depend on $\mathbf{W}_0$, and open up interesting applications in transfer and meta learning, for instance.
> > >
> > > Given that we addressed all of your other concerns raised in the review, we hope that you will consider adjusting your review score while taking the above into account.
> > >
> > > [1] Yao, Yuan, Lorenzo Rosasco, and Andrea Caponnetto. "On early stopping in gradient descent learning." Constructive Approximation 26.2 (2007): 289-315.

---

### Official Review · Reviewer_Rrma · 2021-07-24

**Rating:** 5
**Confidence:** 3

**Summary:**

This paper presents a bound on the expected risk of a two-layer neural network with smooth activation, where the first layer is trained by gradient descent. The bound is in terms of the smallest L2 distance of the weights to initialization among interpolating solutions. When choosing the total number of iterations to be $n^\alpha$ ($\alpha \in(0,1]$), and the width to be bigger than $n^{5\alpha}$, the bound essentially scales like $n^{-\alpha}$ times the squared L2 distance. The analysis is based on stability, and relies on controlling the smallest eigenvalue of the Hessian of the loss.

**Limitations And Societal Impact:**

The discussion is mostly adequate. See main review for some additional questions to be addressed.

**Main Review:**

While much of the stability-based generalization analysis is for convex cases, the current paper is able to obtain a nontrivial result for two-layer neural networks, including the optimization part. I think the result is generally interesting, because there are few results that can guarantee optimization and generalization together. One representative set of such results are based on the NTK, but the current paper uses a different approach and might allow the network to escape the NTK regime.

That said, a few things are unclear to me:
1. How does the bound in this paper compare with the line of work that uses distance to initialization to bound the generalization gap, e.g. [1] below? (BTW, this line of papers are not cited in the current paper.) Admittedly, these papers typically don't have optimization guarantees, but it would still be good to have a careful comparison between the bounds.
2. In the theorem where the NTK bound is recovered from the result in this paper (Theorem 2), why can you assume $\lambda_{\min}(K) \gtrsim 1/n$? This is not assumed in the paper of Arora et al. Furthermore, the result in [Arora et al. 2019] does not require bounded or smooth activation.
3. It would be good to have some discussion about the tightness of the bound. For example, in Lemma 2 and Lemma 3, spectral norm is relaxed to Frobenius norm, which seems loose.
4. One main point the paper stresses is that the analysis is not based on the NTK. However, the result does require the width to grow polynomially with the number of iterations, and it's possible that the network still operates in or close to the NTK regime. Is there an explicit nontrivial theoretical example where the current paper gives an explicit nontrivial bound while NTK cannot?

[1] The role of over-parametrization in generalization of neural networks. Behnam Neyshabur, Zhiyuan Li, Srinadh Bhojanapalli, Yann LeCun, Nathan Srebro. ICLR 2019.


The writing in this paper needs significant improvement. There are many small errors and typos throughout the paper. I'd suggest proofreading the entire paper carefully. Some examples I noticed:
- Inconsistency of conditioning: at some places, $(W_0, u)$ are conditioned on, while at other places only $W_0$ is conditioned on.
- Line 20: should be $|y_i| \le C_y$, not $=$.
- Theorem 2: what does $\tilde{O}_P$ mean? Also, $K$ is never defined.
- Assumption 2: you can't assume the loss is uniformly bounded since this is square loss. It seems that what you really need is that the initial loss $L_S(W_0)$ to be bounded by a constant $C_0$. Is this correct, or am i missing something?
- Assumption 1: there should be absolute values on the 0th, 1st, and 2nd derivatives.
- Line 450: the definition of $\ell$ here is for a linear model...
- Line 477, second step: it should be $\sqrt{L_S(W_s)}$, instead of ${L_S(W_s)}$.
- Line 120: "meaning theat"
- Line 160: "a random inputs"
- Line 161: "use our bounds to we show that"
- Line 164: "recall that that"
- Line 198: "$\phi(u)$ continuous" --> "$\phi(u)$ is continuous"
- ...

**Time Spent Reviewing:**

7

---

> ### Author Response · Authors · 2021-08-09
> **Comment to Reviewer One**
>
> We provide answers to specific questions in the following bullet point format.
>
> * Comparison to the NTK
>     - See comment **Comparision to Neural Tangent Kernel (NTK)** in Comment to all Reviewers.
>
> * Comparison to ``The role of over-parametrization in generalization of neural networks'' by Neyshabur et. al. ICLR 2019
>     - We thank the reviewer for bringing this work to our attention. Indeed, the paper provides generalisation guarantees in term of the norm of the first layer from initialisation. However, the paper works with algorithm-independent uniform-convergence style bounds, and whether gradient descent can decrease the empirical risk within this ball around the initialisation is left out of the picture. In contrast, in our paper we specifically investigate generalisation ability (and stability) of GD, and provide guarantees for both the generalisation and optimisation errors. This reference and discussion is now included within the manuscript.
>
> * Assumption $\lambda_{\min}(\mathbf{K}) \gtrsim 1/n$ and comparision to Arora et. al.
>     - As mentioned within the introduction, it is proven within [Bartlett et al., 2021] that $\lambda_{\min}(\mathbf{K}) \gtrsim d/n$  when using sub-Gaussian inputs (see Lemma 5.3 equation (96) within), and so this holds under assumptions of Arora et al. as well. It is trivial to modify our proof to drop this assumption and to match assumptions of Arora et al. In this case, as in (Arora et al.), the width will depend on the unnormalized eigenvalue $\lambda_{\min}(n \mathbf{K})$.
>
> * Relaxing bounded activation assumption:
>     - Please see a Comment to all Reviewers.
>
> * Extending to non-smooth activations:
>     - Please see a Comment to all Reviewers.
>
> * Tightness of the bound - relaxing Spectral norm to Frobenius norm
>     - While studying the stability of the parameter matrices in spectral norm ought to be tighter, it is currently not clear how the stability analysis of gradient descent should be peformed in norms other than Euclidean (Frobenius for matrices). Regarding tightness, we note in the noiseless case relaxing our upper bounds recovers the bound from [Arora et. al. 2019], and thus, are tighter.
>
> * Typos and writing
>     - We thank the reviewer for highlighting the typos and grammatical errors. These have been fixed and steps have been made to improve the clarity of writing. The notation $\widetilde{O}_{P}(\cdot)$ extends big-Oh notation (up to log-terms) to the stochastic boundedness, that is at least with a certain probability (see for instance https://en.wikipedia.org/wiki/Big_O_in_probability_notation). Indeed, in assumption 2 the constant $C_0$ holds in the loss at initialisation. These two points are clarified within the manuscript.

---

> > ### Comment · Reviewer_Rrma · 2021-08-28
> > **response from reviewer**
> >
> > Thanks to the authors for the response.
> >
> > Regarding comparison/relation to NTK: I understand that the proof in the current paper doesn't rely on the NTK and I found that interesting. A question I asked in the review was "Is there an explicit nontrivial theoretical example where the current paper gives an explicit nontrivial bound while NTK cannot?" It seems that this question is still not answered and thus that the meaningfulness of the result beyond the NTK regime is not entirely clear to me. Therefore I would like to keep my score for now.

---

> > > ### Author Response · Authors · 2021-08-28
> > > **About cases where the current paper gives an explicit nontrivial bound while NTK cannot**
> > >
> > >
> > > Thank you for your feedback! Apologies for not being specific regarding the "*Is there an explicit nontrivial theoretical example where the current paper gives an explicit nontrivial bound while NTK cannot?*"
> > >
> > > Here are some clarifications:
> > >
> > > + We show a trade-off between overparameterisation and convergence rate, $n^{5 \alpha}$ vs $n^{-\alpha}$ respectively. We are not aware of any NTK results capable of this flexibility. One can choose $\alpha$ small enough to break a given polynomial overparameterisation assumption of some NTK result (e.g. Arora et al.) and still have a convergence in our case.
> > >
> > > + We show consistency in the presence of label noise. We are not aware of any published NTK results showing this in a parametric setting. Note that our bounds *do not* introduce any unknown quantities nor require tuning with respect to them to achieve this: To show this in NTK, one could exploit early stopping results in kernel literature such as [1], however such bounds would depend on additional quantities such as spectral profile of a kernel function.
> > >
> > > + Our proof does not require randomised initialisation, which is critical for NTK. Our bounds are therefore strictly more general, depend on $\mathbf{W}_0$, and open up interesting applications in transfer and meta learning, for instance.
> > >
> > > Given that we addressed all of your other concerns raised in the review, we hope that you will consider adjusting your review score while taking the above into account.
> > >
> > >
> > >
> > > [1] Yao, Yuan, Lorenzo Rosasco, and Andrea Caponnetto. "On early stopping in gradient descent learning." Constructive Approximation 26.2 (2007): 289-315.

---

> > > > ### Comment · Reviewer_Rrma · 2021-09-09
> > > > **response**
> > > >
> > > > I don't think these arguments adequately address the comparison with NTK results. While there are several possible generalities in your result as you described, it's possible that the bounds become vacuous in regimes beyond NTK (because it's unclear how $||W^* - W_0||_F^2$ compares with $n^\alpha$). This is why I was repeatedly asking for concrete evidence that you can do better than NTK.

---

> > > > > ### Author Response · Authors · 2021-09-10
> > > > > **Response to Reviewer Rrma**
> > > > >
> > > > > Thank you for your reply!
> > > > >
> > > > > Note that $\mathbf{W}^{\star} \in \arg\min \mathcal{L}(\mathbf{W})$ and so $\|\mathbf{W}^{\star} - \mathbf{W}_0\|_F$ cannot depend on $n$ if data are generated by a *parametric model* (fixed number of parameters) with $\mathbf{W}_0$ chosen such that $\|\mathbf{W}_0\|_F$ is constant (which can be enforced easily).
> > > > >
> > > > > (Arora et al. 2019) consider several such models (without label noise) in Section 6: For example, say that $y_i = \text{cos}(\mathbf{w}^{\star \top} \mathbf{x}_i) - 1$, and so $\mathbf{W}^{\star} = [\mathbf{w}^{\star}, \mathbf{0}, \ldots, \mathbf{0}]$. In our case we have the same with the addition of label noise, whereas (Arora et al. 2019) does not consider such a scenario. This therefore already offers something beyond NTK.
> > > > >
> > > > > Let us know if anything else requires clarification.

---

> > > > > > ### Comment · Reviewer_Rrma · 2021-09-10
> > > > > > **response**
> > > > > >
> > > > > > I don't see why what you wrote is correct. Since the second layer weights are $\pm 1/\sqrt{m}$, something like $W^* = [w^*, 0, \ldots, 0]$ only gives an output scale $O(1/\sqrt{m})$ and thus cannot be an interpolating solution. I'm also not sure it's straightforward to prove that $\|W^*-W_0\|_F$ is a constant. If you believe it's true, please provide a complete proof.
> > > > > >
> > > > > > Re label noise: That [Arora et al. 2019] didn't cover label noise doesn't imply that your results offer something beyond NTK. There are several NTK based papers that can handle label noise via early stopping or regularization (all of which are not cited):
> > > > > > [1] Gradient Descent with Early Stopping is Provably Robust to Label Noise for Overparameterized Neural Networks. M. Li, M. Soltanolkotabi, and S. Oymak
> > > > > > [2] Simple and Effective Regularization Methods for Training on Noisily Labeled Data with Generalization Guarantee. W. Hu, Z. Li, D. Yu
> > > > > > [3] Distillation ≈ Early Stopping? Harvesting Dark Knowledge Utilizing Anisotropic Information Retrieval For Overparameterized Neural Network. B. Dong, J. Hou, Y. Lu, Z. Zhang

---

> > > > > > > ### Author Response · Authors · 2021-09-11
> > > > > > > **Response to Reviewer Rrma**
> > > > > > >
> > > > > > > Thank you a lot for your feedback and clarifying this, which will definitely help to improve the paper!
> > > > > > >
> > > > > > > You're correct, there is a scaling issue in the previous example. On the other hand, regarding when the bound is non-vacuous beyond the NTK regime, note we can have the data generated from a teacher network with weights that have constant norm i.e. $\|\mathbf{W}\|_F^2 = O(1)$. We then have $\|\mathbf{W} - \mathbf{W}_0\|_F^2 = O(\|\mathbf{W}_0\|_F^2)$ which is small provided the initialisation is appropriate. Interestingly, we can learn "simpler" networks with smaller overparameterisation (again, at expense of the rate), while in the NTK setting, overparameterisation would still need to be high to ensure the convergence of the empirical risk. Plus, we still do not require a randomised initialisation, which is required for the NTK analysis.
> > > > > > >
> > > > > > > Regarding the references, indeed these works consider samples corrupted by noise. However, they are only tangentially related to our work (settings are different). Specifically, we focus on early stopping *without* the presence of explicit regularisation, where as reference [2] considers a regularised objective. Meanwhile, [1,3] consider classifications problems in a *low-noise* setting i.e. the noise $\epsilon_0$ is on the order of $O(1/\sqrt{n})$ ([1] assumes $\epsilon_0$ is bounded by the smallest eigenvalue of the empirical kernel matrix which is on this order). In contrast, we consider a real valued response with a general noise level that is *not* required to decrease $n$. Thank you, this discussion, as well as these references are now included within the manuscript.

---

> > > > > > > > ### Comment · Reviewer_Rrma · 2021-09-13
> > > > > > > > **didn't get it**
> > > > > > > >
> > > > > > > > Could you please explain why you can assume $\|W\|_F = O(1)$? Because of the $1/\sqrt{m}$ scaling you would probably need $\|W\|_F = \Omega(\sqrt m)$ to express a fixed function. And your result requires $m$ to grow with $n$.

---

> > > > > > > > > ### Author Response · Authors · 2021-09-13
> > > > > > > > > **Response to Rrma**
> > > > > > > > >
> > > > > > > > > Thank you for your comment.
> > > > > > > > >
> > > > > > > > > The scaling can yield a network whose output remains $O(1)$ as both: the activation can grow linearly at 0 e.g. hyperbolic tangent $\text{tanh}$, Swish or smoothed RELU; and the output consists of summing over the $m$ neurons.
> > > > > > > > >
> > > > > > > > > Specifically, suppose each neuron weight $w_i$ for $i=1,\dots,m$ has norm $| w_i |_2 = O(1/\sqrt{m})$; then each neuron is on the order of $|\phi(\langle w_i,x\rangle)| = O(1/\sqrt{m})$.
> > > > > > > > >
> > > > > > > > > Summing over $m$ neurons and rescaling by $1/\sqrt{m}$ gives an output $\frac{1}{\sqrt{m}} \sum_{i=1}^{m} v_i \phi(\langle w_i,x \rangle) = O(\frac{1}{\sqrt{m}} m \frac{1}{\sqrt{m}}) = O(1)$. Note the norm of the weights satisfy $|W|_2^2 = \sum_i | w_i |_2^2 = O(1)$.
> > > > > > > > >
> > > > > > > > > Please let us know if there is anything else we can clarify.

---

### Author Response · Authors · 2021-08-09
**Comment to all Reviewers**

We thank the reviewers for their feedback. We first provide a general response to some common questions and then address each of reviewers' concerns separately.

$\textbf{Comparison to Neural Tangent Kernel (NTK).}$ Some reviewers cast doubt that our proof goes beyond the NTK regime. Our analysis is different from the NTK because:
* The NTK analysis relies on establishing a connection between a neural net and a kernelized predictor as follows: linearization of the neural network around initialisation $\rightarrow$ random features $\rightarrow$ kernelized predictor. In contrast, **we do not need any of the steps above**, our **analysis is direct**, and **has nothing to do with kernels**.
* Randomization of the initialisation $W_0$ is $\textit{not}$ required for our analysis, whereas it is critical in NTK framework (random features).
* Consequently we are $\textit{not}$ restricted to the RKHS of NTK. Indeed by **relaxing our bound** we recover the NTK-type rates, and thus, ours must be tighter.
* We can trade-off parameterization with convergence rate, i.e. have the width scale $m = n^{5\alpha}$ for a statistical rate of $n^{-\alpha}$ for $\alpha \in (0,1]$. To the best of our knowledge, **this is not possible in the NTK-style proofs** for arbitrarily small $\alpha$ without breaking the global convergence.
* Importantly, polynomial parameterization, i.e. $m = \text{poly}(n)$, **does not imply** NTK regime or kernelization, it is simply required for achieving a global convergence.

$\textbf{Extending to non-smooth activations.}$ One of the goals of the paper is to $\textit{simply}$ demonstrate a novel stability analysis without NTK. Using smooth activations removes a lot of tangential complexity, which we believe would only obscure the proof and our main message.

However, one can still achieve this by keeping dependence on the first order Taylor error of loss, as in [Arora et al. 2019], and keeping track of activation patterns throughout training. This error can then be controlled by the initialisation, and is traded-off with the stepsize and iterations $\eta t$ through the optimisation error. Since this analysis is specialised for the non-smooth case, we leave this direction to future work.

Finally, from a practitioner's point of view smooth activations are often superior to ReLU even at the Imagenet scale $[4]$, and as such we do not see a definitive advantage of non-smooth activations, and we believe that smooth activations have their own merit.

$[4]$ Ramachandran, P., Zoph, B., and Le, Q. V. (2017). ``Searching for activation functions'' arXiv:1710.05941

$\textbf{Relaxing bounded activation assumption.}$ The bounded activation assumption allows for simplified proofs, and our results hold (with minor modifications) without this condition.
We can achieve this by localising the smoothness analysis  (using a Taylor approximation around each GD iterate), with the residual at the GD iterate $\mathcal{L}_{S}(\mathbf{W}_t)$ taken  instead of a uniform bound (see equation 12 in the Supp. Material).

---

### Decision · Program_Chairs · 2021-09-27

**Decision:**

Accept (Poster)

**Comment:**

This paper presents a bound on the expected risk of a two-layer neural network with smooth activation, where the first layer is trained by gradient descent. The bounds cover noisy and noiseless cases and depend on the distance of the interpolation solution which is closest to the initialization.

The reviewers agree that the paper provides a theoretical analysis for the generalization gap of two-layer neural networks using algorithmic stability using a proof technique that is not based on the Neural Tangent Kernel, while obtaining results that have a similar generality of the ones of NTK for two layers networks. Moreover, as observed by some reviewers, the bounds require the width of the networks to be bigger than a polynomial of the number of iterations (specifically, number of iterations times the step size), observing that it also works out of the NTK regime.

In any case, according to the reviewers, the paper provides a relatively novel proof strategy in the context of theoretical analysis of the generalization of deep learning that is conceptually simpler, and also able to shed light on the properties of the first non-trivial example of a dnn, i.e. a 2-layer nn. We think that this contribution is enough for publication.

However, we strongly encourage the authors to improve the writing of the paper, in particular Section 3, as observed by some reviewers.